# Dynamic modulation of decision biases by brainstem arousal systems

**Jan Willem de Gee[1,2]\*, Olympia Colizoli[1,2,3], Niels A Kloosterman[2,3,4], Tomas Knapen[5], Sander Nieuwenhuis[6], Tobias H Donner[1,2,3]\***

[1]Department of Neurophysiology and Pathophysiology, University Medical Center Hamburg-Eppendorf, Hamburg, Germany; [2]Department of Psychology, University of Amsterdam, Amsterdam, The Netherlands; [3]Amsterdam Brain & Cognition, University of Amsterdam, Amsterdam, The Netherlands; [4]Max Planck UCL Centre for Computational Psychiatry and Ageing Research, Max Planck Institute for Human Development, Berlin, Germany; [5]Department of Experimental and Applied Psychology, Vrije Universiteit Amsterdam, Amsterdam, The Netherlands; [6]Institute of Psychology, Leiden University, Leiden, The Netherlands

**Abstract** Decision-makers often arrive at different choices when faced with repeated presentations of the same evidence. Variability of behavior is commonly attributed to noise in the brain's decision-making machinery. We hypothesized that phasic responses of brainstem arousal systems are a significant source of this variability. We tracked pupil responses (a proxy of phasic arousal) during sensory-motor decisions in humans, across different sensory modalities and task protocols. Large pupil responses generally predicted a reduction in decision bias. Using fMRI, we showed that the pupil-linked bias reduction was (i) accompanied by a modulation of choice-encoding pattern signals in parietal and prefrontal cortex and (ii) predicted by phasic, pupil-linked responses of a number of neuromodulatory brainstem centers involved in the control of cortical arousal state, including the noradrenergic locus coeruleus. We conclude that phasic arousal suppresses decision bias on a trial-by-trial basis, thus accounting for a significant component of the variability of choice behavior.

\*For correspondence: jwdegee@ gmail.com (JWdG); t.donner@uke. de (THD)

**Competing interests:** The authors declare that no competing interests exist.

## Introduction

Decision-makers often arrive at different choices in the face of repeated presentations of the same evidence (*Glimcher, 2005*; *Gold and Shadlen, 2007*; *Shadlen et al., 1996*; *Sugrue et al., 2005*; *Wyart and Koechlin, 2016*). This intrinsic behavioral variability is typically attributed to spontaneous fluctuations of neural activity in the brain regions computing decisions (*Glimcher, 2005*; *Shadlen et al., 1996*) (but see [*Beck et al., 2012*; *Brunton et al., 2013*]). Indeed, fluctuations of neural activity are ubiquitous in the cerebral cortex (*Faisal et al., 2008*; *Glimcher, 2005*; *Lin et al., 2015*).

One candidate source of these fluctuations in cortical activity is systematic variation in central arousal state. Central arousal state is controlled by the neuromodulatory systems of the brainstem, which have widespread projections to cortex and tune neuronal parameters governing the operating mode of their cortical target circuits (*Aston-Jones and Cohen, 2005*; *Harris and Thiele, 2011*; *Lee and Dan, 2012*). Importantly, these neuromodulatory systems operate at different timescales (*Aston-Jones and Cohen, 2005*; *Parikh et al., 2007*). Some, in particular the noradrenergic locus coeruleus (LC), are rapidly recruited, in a time-locked fashion, during elementary decisions (*Aston-Jones and Cohen, 2005*; *Bouret and Sara, 2005*; *Dayan and Yu, 2006*; *Parikh et al., 2007*). Pupil diameter, a reliable peripheral marker of central (cortical) arousal state (*McGinley et al., 2015b*),

**eLife digest** When asked to make repeated decisions we will often choose differently each time even when we are given the same information to inform our choice. A stock trader, for example, will typically be more inclined to buy on some days and sell on others even if the financial markets remain unchanged. Fluctuations in the brain's level of alertness or excitability, otherwise known as its arousal, are thought to contribute to this variability in decision-making.

An area at the base of the brain called the brainstem – and in particular one of its subregions, the locus coeruleus – helps shape arousal levels by releasing chemicals called neuromodulators. For reasons that remain unknown, activation of the locus coeruleus also causes the pupil of the eye to suddenly increase in size. Now, de Gee et al. have exploited this link to unravel how changes in brain arousal lead to systematic changes in decision-making.

Volunteers were asked to judge whether a faint pattern was embedded in flickering noise on a computer screen, and to report their judgment by pressing one of two buttons to indicate "yes" or "no". Although the decision was comparatively simple, it did involve evaluating changing information over time before making a choice – like when considering the stock market. As the volunteers performed the task, de Gee et al. measured their brain activity and the size of their pupils. Most of the volunteers had a tendency to respond "no" even when the pattern was present. However, whenever their locus coeruleus was particularly active, and their pupils increased in size, their decision process was changed so that this unhelpful choice bias decreased.

This suggests that by boosting arousal, the locus coeruleus reduces existing biases in our decision-making. Varying levels of locus coeruleus activity may thus explain why we can reach different conclusions when considering the same information on multiple occasions. The next challenge is to identify what it is about the decision-making process that activates the locus coeruleus on some occasions but not others.

also increases during decisions (*Beatty, 1982*; *de Gee et al., 2014*; *Gilzenrat et al., 2010*; *Lempert et al., 2015*; *Nassar et al., 2012*). These observations point to an important role of phasic (i.e., fast) pupil-linked arousal signals in decision-making (*Aston-Jones and Cohen, 2005*; *Dayan and Yu, 2006*). Yet, the precise nature of this role has remained unknown.

Here, we investigated how phasic, task-related arousal interacts with decision computations in the human brain. We combined pupillometry, fMRI, and computational modeling to probe into the interplay between task-related arousal and decision computations underlying elementary sensory-motor choice tasks. Sensory-motor decisions entail the gradual accumulation of noisy 'sensory evidence' about the state of the world towards categorical decision states governing behavioral choice (*Bogacz et al., 2006*; *Brody and Hanks, 2016*; *Gold and Shadlen, 2007*; *Ratcliff and McKoon, 2008*). A large-scale network of regions in frontal and parietal cortex seems to accumulate stimulus responses provided by sensory cortices towards choices of motor movements (*Gold and Shadlen, 2007*; *Siegel et al., 2011*) (but see [*Brody and Hanks, 2016*; *Katz et al., 2016*]). We here aimed to elucidate the interaction between pupil-linked arousal responses, evidence accumulation, and decision processing across several (cortical and subcortical) brain regions.

Large task-evoked pupil responses were consistently accompanied by a reduction in perceptual decision bias in different sensory modalities (visual and auditory) and task protocols (detection and discrimination). Decision bias reflects the degree to which an observer's choice deviates from the objective sensory evidence. Using fMRI for one of these tasks revealed that the bias reduction was accompanied by a modulation of choice-encoding pattern signals in prefrontal and parietal cortex. Further, the bias reduction was predicted by task-evoked, pupil-linked responses in a network of neuromodulatory brainstem nuclei controlling cortical arousal state. We conclude that phasic neuromodulatory signals reduce biases in the brain's decision-making machinery. As a consequence, phasic arousal accounts for a significant component of the variability of choice behavior, over and above the objective evidence gathered from the outside world.

## Results

We systematically quantified the interaction between pupil-linked arousal responses and decision computations at the algorithmic and neural levels of analysis. We here operationalize 'phasic arousal' as task-evoked pupil responses (TPR). This operational definition is based on recent animal work, which established remarkably strong correlations between non-luminance mediated variations in pupil diameter and global cortical arousal state (*McGinley et al., 2015b*).

The Results section is organized as follows. First, we quantify TPRs during the main behavioral task studied in this paper. The key observation here was the substantial trial-to-trial variability of the TPR amplitude. All subsequent analyses exploited this variability to pinpoint the functional correlates of phasic arousal. We then present results from modeling TPR-dependent changes in choice behavior, identifying precise algorithmic correlates of phasic arousal. These results yielded detailed predictions for the underlying modulations of cortical signals. Third, we present tests of these predictions, focusing on functionally delineated cortical regions of interest. We conclude by establishing that the trial-to-trial fluctuations in TPR amplitude, and the associated bias reduction, were closely linked to task-evoked responses of neuromodulatory brainstem centers involved in regulating cortical arousal state.

### Tracking trial-to-trial fluctuations in phasic arousal

The main task used in this study was detection ('yes-no', simple forced choice protocol) of a low-contrast grating (*Figure 1A*). The grating contrast was titrated to the 75% correct level, and subjects did not receive trial-by-trial feedback. As observed previously (*de Gee et al., 2014*), TPR amplitudes during this task fluctuated widely from trial to trial (*Figure 1B,C*; see Materials and Methods for quantification of TPR). To illustrate, pooling trials into two bins containing the lowest and highest 40% of TPR amplitudes (*Figure 1B*) yielded, on average, the commonly observed task-evoked pupil dilations for the high TPR bin, but pupil constrictions for the low TPR bin (*Figure 1C*). We used a previously established model to estimate the time course of the neural input driving the measured TPRs (GLM; see Materials and methods; *Figure 1—figure supplement 1A–C*). This revealed that the difference between the low and high TPR bins was primarily due to the difference in a sustained component that spanned the entire interval from cue to behavioral choice (*Figure 1D*). The difference of the sustained component between low and high TPR was significantly larger than the corresponding difference for two components at cue or choice, respectively (2-way repeated measures ANOVA with factors temporal component and TPR bin; interaction: $F_{2,26} = 79.00$, $p<0.001$).

In sum, TPR amplitude exhibited substantial trial-to-trial fluctuations, which were predominantly driven by changing levels in sustained input during decision formation. Given the prolonged nature of the decision (median of subject-median reaction time, RT: 2.11 s), the sustained, intra-decisional arousal boost might have interacted with the decision computation. To test for such an interaction between arousal boost and decision computation, we next modeled subjects' choice behavior as a function of TPR amplitude.

### Phasic arousal is inversely related to decision bias

We found a robust and consistent relationship between TPR and decision bias. This effect was present in two independent data sets using an analogous contrast detection task: the newly collected fMRI data set, and a re-analysis of an existing data set (*de Gee et al., 2014*)) (*Figure 2A,D*, middle and right panels). Decision bias was quantified in two ways (for details, see Materials and methods). First, we computed signal detection-theoretic (SDT) criterion (*Figure 2A,D*, middle panels). Second, we computed the fraction of 'yes'-choices (right panels), after balancing the number of signal+noise and noise trials within each TPR bin. We did not find a consistent relationship between phasic arousal, as measured by TPR, and perceptual sensitivity, quantified by SDT d' (*Figure 2A,D*, left panels).

The negative association between TPR and decision bias (SDT criterion) was approximately linear across a range of five TPR-defined bins (*Figure 2B,E*, right panels). In all cases, here and below, we tested whether fits of second-order polynomials, reflecting non-monotonic relationships between TPR and behavior, were superior to the linear fits (via sequential polynomial regression analysis; Materials and methods). We found a non-monotonic relationship between TPR and sensitivity in the behavioral data set from *de Gee et al. (2014)*, but not in the fMRI dataset (*Figure 2B,E*, left panels).

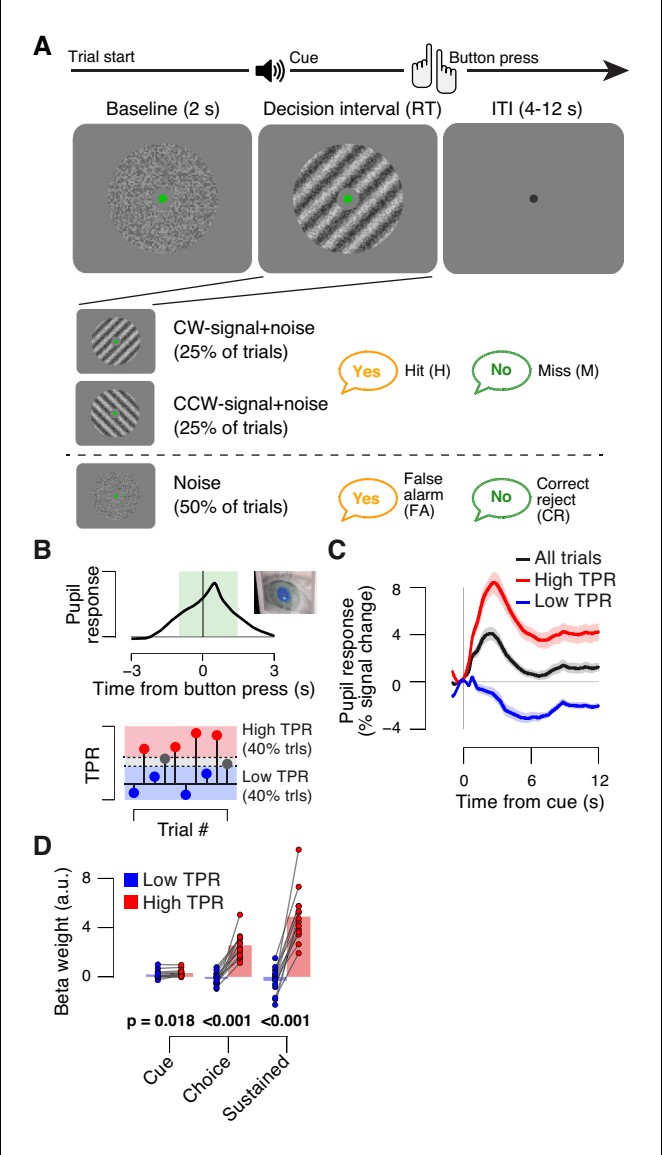

**Figure 1.** Behavioral task and task-evoked pupil responses. (**A**) Yes-no contrast detection task. Top: schematic sequence of events during a signal+noise trial. Subjects reported the presence or absence of a faint grating signal superimposed onto dynamic noise. Bottom left: the signal, if present, was oriented clockwise or counter clockwise on different blocks (known to the subject beforehand). Signal contrast is high for illustration only. Bottom right: trial types. (**B**) Quantifying task-evoked pupillary response (TPR) amplitude. Top: mean TPR time course of an example subject. Green box, interval for averaging TPR values on single trials. Bottom: trials were pooled into three bins of TPR amplitudes (lowest/highest 40% and intermediate 20%). (**C**) TPR time course for the three bins. (**D**) Mean beta weights of transient (cue, choice) and sustained input components under low vs. high TPR, estimated with a general linear model (see Materials and methods; *Figure 1—figure supplement 1A,B*), separately for low and high TPR trials. Panels C, D: group average (N = 14); shading, s.e.m.; data points, individual subjects; stats, permutation test.

The following figure supplement is available for figure 1:

**Figure supplement 1.** Linear modeling of TPR.

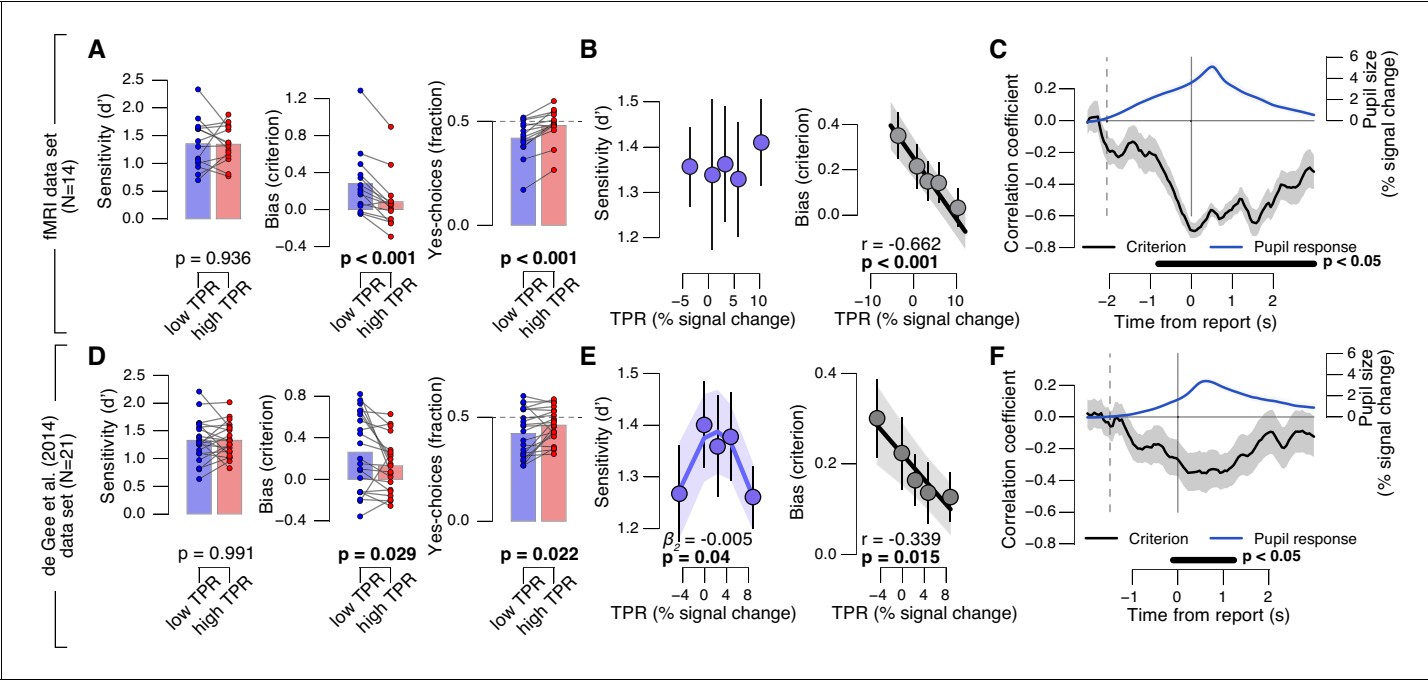

**Figure 2.** Phasic arousal predicts reduction of choice bias. (A) Perceptual sensitivity SDT d' (left), decision bias, measured as SDT criterion (middle) or fraction of 'yes'-choices (right), for low and high TPR. For the fraction of 'yes'-choices analysis, we ensured that each TPR bin consisted of an equal number of signal+noise and noise trials (see Materials and methods). Data points, individual subjects. (B) Relationship between TPR and d' or criterion (5 bins). Linear fits are plotted wherever the first-order fit was superior to the constant fit (see Materials and methods). Quadratic fits are plotted wherever the second-order fit was superior to first-order fit. (C) Sliding window linear correlation between TPR and SDT criterion (5 bins), aligned to button press. Dashed line, median decision onset (cue). The group average pupil response time course is plotted for reference in blue. (D–F) As panels A-C, for an independent data set (*de Gee et al., 2014*). All panels: group average (N = 14 and N = 21); shading or error bars, s.e.m.; stats, permutation test.

The following source data and figure supplement are available for figure 2:

**Source data 3.** Table with variable identifiers used in *Figure 2—source data 1* and *2*.
**Source data 1.** This csv table contains the data for *Figure 2* panel A.
**Source data 2.** This csv table contains the data for *Figure 2* panel D.
**Figure supplement 1.** Phasic arousal predicts reduction of choice bias.

This non-monotonic (inverted U-shape) relationship between pupil diameter and sensitivity is consistent with previous animal work on correlations between baseline arousal and behavior (*Aston-Jones and Cohen, 2005*; *McGinley et al., 2015a*). However, it was less consistent across the data sets analyzed in this paper than the negative linear effect of TPR on decision bias. The consistent effect of TPR on decision bias has not been reported before in previous studies of slow fluctuations of baseline pupil diameter. In what follows, we focus on the negative effect of TPR on decision bias.

Most subjects were overall (i.e., without splitting trials by TPR) intrinsically biased to respond 'no': 10 out of 14 subjects exhibited a significantly conservative criterion (within-subject permutation tests; p<0.05) in the fMRI data set, and 14 out of 21 subjects in the data set from *de Gee et al. (2014)*. Because signal+noise and noise trials were equally frequent in both experiments, this bias was always maladaptive. Critically, this maladaptive bias was particularly pronounced under low TPR; but under high TPR the bias was nearly neutralized, especially in the fMRI data set (criterion around zero, and fraction of 'yes'-choices around 0.5 for highest TPR bins, *Figure 2A,B*).

## A robust effect of phasic arousal on the decision computation

A number of control analyses and experiments supported the idea that the negative correlation between TPR amplitude and decision bias reflected a specific effect of phasic arousal on the decision computation that generalized across perceptual choice tasks. First, the effect emerged during, not after, decision formation: a sliding-window correlation between TPR and criterion became negative from decision onset onwards, and reached statistical significance before button press (*Figure 2C,F*). In the fMRI data set, this correlation was highly significant more than 800 ms before button press (*Figure 2C*). Given the sluggish nature of the pupil response (see above), the underlying central arousal transients must have occurred even earlier than that, leaving substantial time for shaping the decision outcome.

Second, there was no robust association between baseline pupil diameter and decision bias (*Figure 2—figure supplement 1A–D*). This ruled out possible concerns that the effect might be due to corresponding (opposite) associations between baseline pupil diameter and behavior, 'inherited' by TPR through its negative correlation with baseline pupil diameter (*de Gee et al., 2014*).

Third, the effect of TPR on decision bias was robust with respect to the details of the analysis approach. For *Figure 2*, as for all other analyses reported in the main text, we removed (via linear regression) components explained by RT. The rationale was to specifically isolate variations in the amplitudes of the neural responses driving TPR, irrespective of RT, variations of which might also cause variations of TPR amplitude without changes in the underlying neural response amplitudes (for details see Materials and methods). We observed the same linear effect of TPR on bias without removing trial-to-trial variations in TPR that were due to RT (*Figure 2—figure supplement 1E–J*).

## Pupil-linked bias reduction is a general phenomenon

Fourth, the effect of TPR on decision bias shown in *Figure 2* generalized to other perceptual choice tasks, which differed on several dimensions from the main contrast detection task used in this paper (*Figure 3*). In one follow-up experiment, we measured pupil-linked behavior during an auditory yes-no (tone-in-noise) detection task near psychophysical threshold using the same stimuli as in (*McGinley et al., 2015a*) (see Materials and methods). The only visual stimulus was a stable fixation dot. The decision interval contained only auditory noise (the same as in (*McGinley et al., 2015a*)) on half the trials, and a pure sine wave superimposed onto the noise on the other half of the trials. Again, TPR predicted a significant (linear) reduction in conservative decision bias, and an increased tendency to respond 'yes' (*Figure 3A,B*). TPR also exhibited a non-monotonic relationship with sensitivity, as observed in rodents for baseline pupil diameter in (*McGinley et al., 2015a*).

Another follow-up experiment assessed whether the pupil-linked bias reduction observed above may have been due to the asymmetric nature of the detection tasks (i.e., discriminating the presence from the absence of a signal) or due to the absence of single-trial feedback. Symmetric two-alternative forced choice tasks are commonly associated with weaker biases than yes-no detection tasks (*Green and Swets, 1966*). We used a symmetric visual random dot motion (up vs. down) discrimination task near psychophysical threshold with feedback after each trial (see Materials and methods). Although many subjects exhibited clear biases for reporting one or the other direction, these were more evenly distributed around zero than in the above yes-no tasks, in which the sign of the bias was largely consistent across individuals. Therefore, we here analyzed subjects' absolute criterion values (i.e., overall bias regardless of sign) and fraction of non-preferred choices (i.e., the choice opposite to their general bias, irrespective of TPR). Again, TPR predicted a reduction in absolute decision bias, and an increase in the fraction of non-preferred choices (*Figure 3C,D*), analogous to the effects observed for the detection tasks above.

In sum, a number of analyses and experiments showed that pupil-linked, phasic arousal was consistently associated with a monotonic reduction in perceptual decision biases in different sensory modalities and task protocols.

## Phasic arousal predicts a reduction of evidence accumulation bias

To further pinpoint the nature of the TPR-induced bias suppression, we fitted the drift diffusion model, an established dynamic model of two-choice decision processes (*Figure 4A*; [*Ratcliff and McKoon, 2008*]) to subjects' RT distributions from the main task (contrast detection). The drift

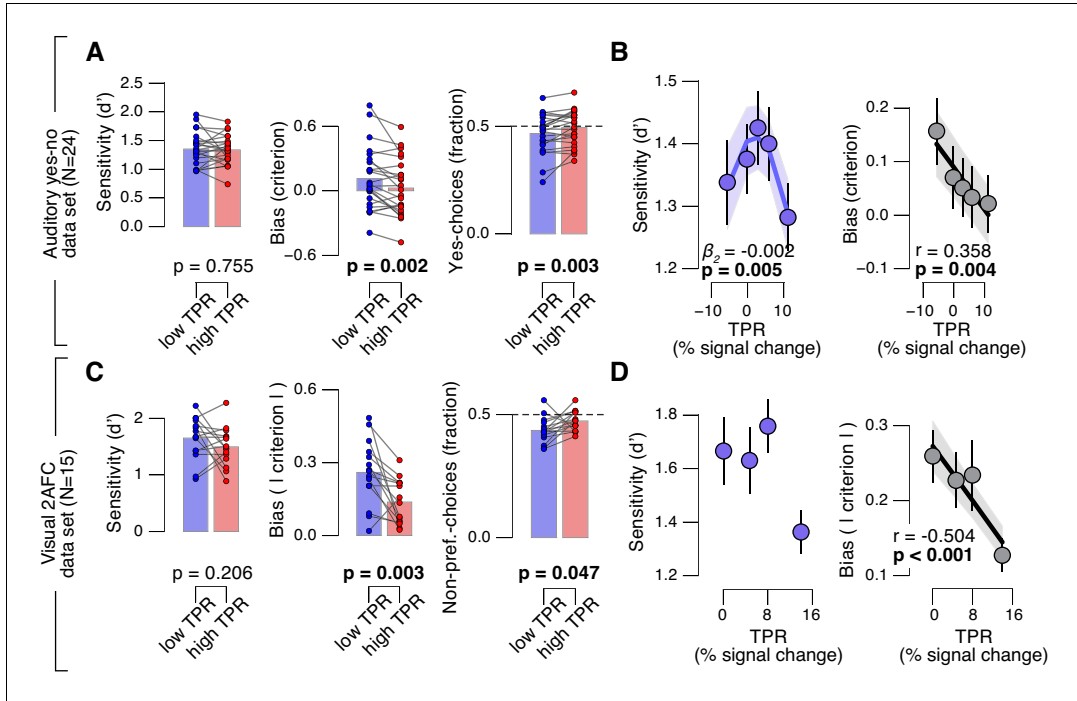

**Figure 3.** Arousal-linked bias reduction generalizes to other choice tasks. (**A**) Perceptual sensitivity (d'; left) and decision bias, measured as criterion (middle) or fraction of 'yes'-choices (computed as for *Figure 2A*, right), for low and high TPR. Data points, individual subjects. (**B**) Relationship between TPR and d' or criterion (5 bins). Linear fits were plotted wherever the first-order fit was superior to the constant fit (see Materials and methods). Quadratic fits were plotted wherever the second-order fit was superior to first-order fit. (**C**) Perceptual sensitivity (d', left) and decision bias, measured as absolute criterion (middle) or fraction of non-preferred choices (right), for low and high TPR. For the fraction of non-preferred choices analysis, we ensured that each TPR bin consisted of an equal number of motion up and down trials (see Materials and methods). (**D**) Relationship between TPR and d' or absolute criterion (4 bins instead of 5, because of fewer trials per subject, see Materials and methods). All panels: group average (N = 24 and N = 15); shading or error bars, s.e.m.; stats, permutation test.

The following source data is available for figure 3:

**Source data 3.** Table with variable identifiers used in *Figure 3—source data 1* and *2*.
**Source data 1.** This csv table contains the data for *Figure 3* panel A.
**Source data 2.** This csv table contains the data for *Figure 3* panel C.

diffusion model posits the perfect accumulation of noisy sensory evidence towards one of two decision bounds, here for 'yes' and 'no' (*Figure 4A*).

We fitted the model separately for low and high TPR trials (see *Figure 4B* for an individual example). Within the model, the TPR-induced reduction of conservative bias, evident in *Figures 2* and *3*, may have been brought about by two distinct mechanistic scenarios: (i) the evidence accumulation process started from a level closer to the 'yes'-bound (i.e., a change in the 'starting point' parameter); or (ii) the accumulation process was driven more towards the 'yes'-bound (i.e., a change in the 'drift criterion' parameter). The drift criterion is equivalent to an evidence-independent constant added to the drift. A non-zero drift criterion results in a bias of the decision variable that grows linearly with time. Although clearly distinct in nature, both mechanisms (starting point and drift criterion) would have resulted in an increase in the fraction of 'yes'-choices, and thus a reduction of decision bias. Critically, both mechanisms were distinguishable through their distinct effects on the shape of the RT distribution (*Figure 4—figure supplement 1*). To dissociate between these alternative

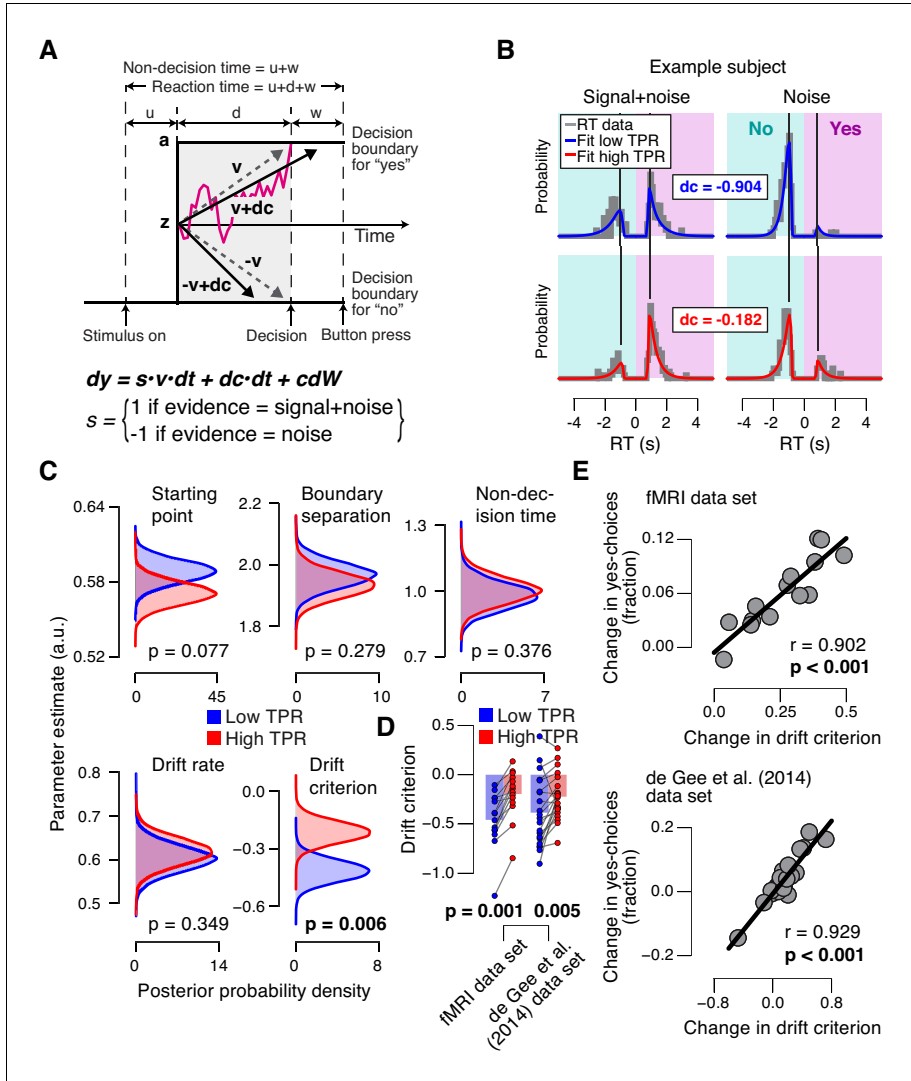

**Figure 4.** Phasic arousal predicts reduction of accumulation bias. (**A**) Schematic and simplified equation of drift diffusion model accounting for RT distributions for 'yes'- and 'no'-choices ('stimulus coding'; see Materials and methods). Notation: *dy*, change in decision variable y per unit time *dt*; *v•dt*, mean drift (multiplied with 1 for signal +noise trials, and −1 for noise trials); *dc•dt*, drift criterion (an evidence-independent constant added to the drift); and *cdW*, Gaussian white noise (mean = 0, variance = $c^2$ dt). (**B**) RT distributions of one example subject for 'yes'- and 'no'-choices, separately for signal+noise and noise trials and separately for low and high TPR. RTs for 'no'-choices were sign-flipped for illustration purposes. Straight lines, mode (i.e., maximum) of the fitted RT distributions. Please note that TPR predicts an increased fraction of 'yes'-choices with only a minor change of the mode of the RT distribution, consistent with a drift criterion effect rather than a starting point effect (**Figure 4—figure supplement 1**). (**C**) Group-level posterior probability densities for means of parameters. To maximize the robustness of parameter estimates (**Wiecki et al., 2013**), two data sets were fit jointly (the current fMRI and our previous study (**de Gee et al., 2014**); N = 35). Starting point (z) is expressed as a proportion of the boundary separation (a). (**D**) Drift criterion point estimates for low and high TPR trials, separately for both data sets (N = 14 and N = 21, respectively). Data points, individual subjects; stats, permutation test. (**E**) Change in fraction of 'yes'-choices for low vs. high TPR trials, plotted against change in drift criterion. Data points, individual subjects.

The following source data and figure supplements are available for figure 4:

**Source data 2.** Table with variable identifiers used in **Figure 4—source data 1**.

**Source data 1.** This csv table contains the data for **Figure 4** panel D.

*Figure 4 continued*

**Figure supplement 1.** Effects of starting point vs. drift criterion on RT distributions.
**Figure supplement 2.** Phasic arousal predicts reduction of accumulation bias.

mechanisms we fitted the model, while allowing several model parameters (boundary separation, non-decision time, mean drift rate, starting point, and drift criterion) to vary with TPR.

The model fits (see Materials and methods and [*Wiecki et al., 2013*]) supported the second mechanism: a change in drift criterion. An individual example is shown in *Figure 4B*, and group data are shown in *Figure 4C*. Drift criterion was generally negative, indicating an overall conservative accumulation bias towards the bound for 'no'-choices. But drift criterion was pushed closer towards zero under high TPR, indicating an unbiased drift, as optimal for the current task (*Figure 4B,C*). The other main parameters (including starting point and mean drift rate) were not significantly affected by TPR. The TPR-linked effect on drift criterion was also evident in the individual point estimates from the fMRI sample only (*Figure 4D*).

Again, we we found no evidence for an effect on any parameter of the drift diffusion model when comparing trials with low and high baseline pupil diameters (*Figure 4—figure supplement 2A*), and we obtained qualitatively identical results without removing trial-to-trial variations of RT from the TPR amplitudes (*Figure 4—figure supplement 2B–D*; Materials and methods).

As a control of the significance of the TPR-dependent effect on drift criterion, we re-fitted the model, but now fixing drift criterion with TPR, while still allowing all other of the above parameters to vary with TPR. In this variant of the model, we again found no TPR-dependent change in any of the other parameters (boundary separation: p=0.428; non-decision time: p=0.370; starting point: p=0.117; mean drift rate: p=0.361). Critically, model comparison favored the complete version of the model with TPR-dependent variation in drift criterion (deviance information criterion, 50437 vs. 50528, respectively; see Materials and methods). This implies that the TPR-dependent variability in accumulation bias was essential to account for the TPR-dependent effects on behavior.

The individual changes in drift criterion between low vs. high TPR trials established by means of diffusion modeling accounted for a substantial fraction of the individual differences in TPR-predicted changes in the fraction of 'yes'-choices (*Figure 4E*) obtained in the model-free analyses (*Figure 2A, D*, right panels). TPR-related changes in starting point had a weaker, and statistically not significant, effect on the fraction of 'yes'-choices (fMRI data set: r = −0.345, p=0.227; *de Gee et al. (2014)* data set: r = −0.419, p=0.059).

In sum, in the decision task studied here, pupil-linked, phasic arousal predicted a reduction of conservative bias, specifically in the evidence accumulation, and was neither reflected in the baseline level of the decision variable at the start of the accumulation nor its mean drift. In other words, TPR accounted for a portion of the trial-to-trial variability in the drift unrelated to the objective sensory evidence. This correlate of phasic arousal at the algorithmic level was in line with the notion that phasic arousal shapes decision outcome by interacting with the evidence accumulation computation that lies at the heart of the decision process.

Taken together, the behavioral modeling results reported in *Figures 2–4* put strong constraints on the expected changes in cortical decision processing due to phasic arousal. Specifically, changes in the encoding of the incoming evidence by sensory cortical areas, as observed in previous work on fluctuations in baseline arousal levels (*McGinley et al., 2015a*; *Reimer et al., 2014*; *Vinck et al., 2015*), would be associated with changes in perceptual sensitivity. However, we found that TPR was not associated with any robust change in sensitivity (measured as d' or as mean drift rate) in the fMRI dataset, thus, predicting no TPR-linked modulation of sensory responses in visual cortex. Instead, the observed effect of TPR on choice bias (criterion, drift criterion) predicted a directed shift (towards 'yes') in neural signals encoding subjects' choices, in downstream cortical regions. We next tested these predictions by assessing the relationship between TPR and (i) stimulus-specific responses in early visual cortex, and (ii) choice-specific responses in downstream cortical regions.

## Phasic arousal does not boost sensory responses in visual cortex

The fMRI response in early visual cortex (areas V1, V2, and V3) during near-threshold visual tasks is made up of distinct components, including a (weak and focal) stimulus-specific component and a (large and global) task-related, but stimulus-independent, component (*Cardoso et al., 2012*; *Donner et al., 2008*; *Ress et al., 2000*). We used an approach based on multi-voxel pattern analysis analogous to previous work (*Choe et al., 2014*; *Pajani et al., 2015*) to isolate the stimulus-specific response component. Because the majority of visual cortical neurons encoding stimulus contrast are also tuned to stimulus orientation, orientation-tuning could serve as a 'filter' to separate the cortical stimulus response from stimulus-unrelated signals. Specifically, the low contrast signal in our task should have evoked a small response in each visual cortical neuron selective for the orientation of the target signal (45° or 135°, on different experimental runs, *Figure 1A*) across a substantial part of the retinotopic map. Thus, the presence or absence of the target signal should be reliably encoded in the orientation-specific component of the cortical population response, within the retinotopic sub-region corresponding to the signal. We first individually delineated these retinotopic sub-regions within each of V1-V3 (see *Figure 5A* for an example subject) and then quantified the orientation-specific response component therein as the spatial correlation of multi-voxel response patterns with an orientation-specific 'template' (Materials and methods).

As expected, this orientation-specific response component differed robustly between signal +noise and noise trials (*Figure 5B*). A 2-way repeated measures ANOVA with factors stimulus and TPR bin yielded a highly significant main effect of stimulus for V1, V2, and V3 (V1: $F_{1,13} = 303.5$, V2: $F_{1,13} = 646.3$, V3: $F_{1,13} = 316.6$; all p<0.001).

The orientation-specific response component also reliably discriminated between signal+noise and noise trials on a single-trial basis (*Figure 5—figure supplement 1*). Consequently, we henceforth refer to this component as the 'stimulus-specific response'. However, the stimulus-specific response was not boosted under high TPR (*Figure 5B*, no significant main effect of TPR, nor stimulus x TPR interaction in any of V1-V3).

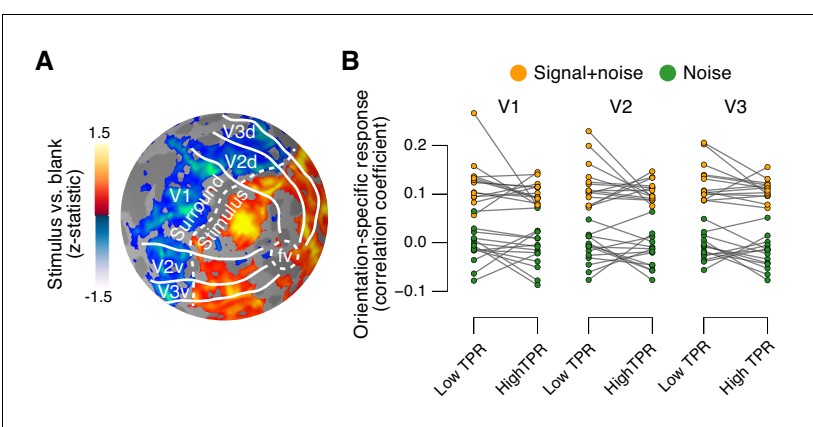

**Figure 5.** Phasic arousal does not boost sensory responses in visual cortex. (**A**) Map of fMRI responses during stimulus localizer runs (see Materials and methods); example subject. V1-V3 borders were defined based on a separate retinotopic mapping session. 'Stimulus sub-regions', regions with positive stimulus-evoked response; 'surround sub-regions', regions with negative stimulus-evoked response. (**B**) Orientation-specific fMRI responses in 'center' sub-regions of V1-V3, separately for signal+noise and noise trials, and separately for low and high TPR trials. Statistical tests are reported in main text. Data points, individual subjects (N = 14); stats in main text.

The following figure supplement is available for figure 5:

**Figure supplement 1.** Quantifying single-trial reliability of stimulus-specific responses.

## No evidence for arousal-dependent boost of sensory responses in any cortical area

The above analysis focused on the stimulus-specific response in early visual cortex. To avoid missing TPR-dependent modulations of sensory responses in higher cortical regions, we also mapped out modulations of fMRI responses by TPR across cortex (see Materials and methods). Various regions including visual, parietal, prefrontal, and motor cortices exhibited robust task-evoked overall fMRI responses (i.e., difference between the decision interval and baseline; *Figure 6A*), as well as robust modulations by TPR (*Figure 6B*), whereby TPR-induced boosts only partly overlapped with the task-positive responses.

However, in no single region did the overall fMRI responses differ between signal+noise and noise trials (*Figure 6C*). This indicates that our multi-voxel pattern approach described above was, in fact, essential for detecting the weak cortical response to the near-threshold target signals. Critically, in no region did we find a significant interaction between the factors stimulus (signal+noise vs. noise) and TPR (low vs. high TPR; *Figure 6D*).

Taken together, both complementary analyses showed that phasic, task-evoked arousal signals did not modulate cortical responses encoding the presence of the low-contrast signal. This is in line with the lack of TPR-linked change in perceptual sensitivity in the fMRI dataset (*Figure 2A*, *Figure 4D*).

## Phasic arousal modulates choice-specific signals in frontal and parietal cortex

We then sought to test for directed shifts in neural signals encoding subjects' choices under high TPR, which would be in line with the changes in decision biases identified by behavioral modeling. Here, we use the term 'choice-specific' to refer to fMRI-signals that reliably discriminated between subjects' choice ('yes' vs. 'no'). Two complementary approaches delineated several cortical regions that exhibited such choice-specific signals (*Figure 7*). The first approach (*Figure 7A*) was based on the lateralization of fMRI responses with respect to the motor effector used to report the choice (i. e., response hand; see (*de Lange et al., 2013*; *Donner et al., 2009*) and Materials and methods). In

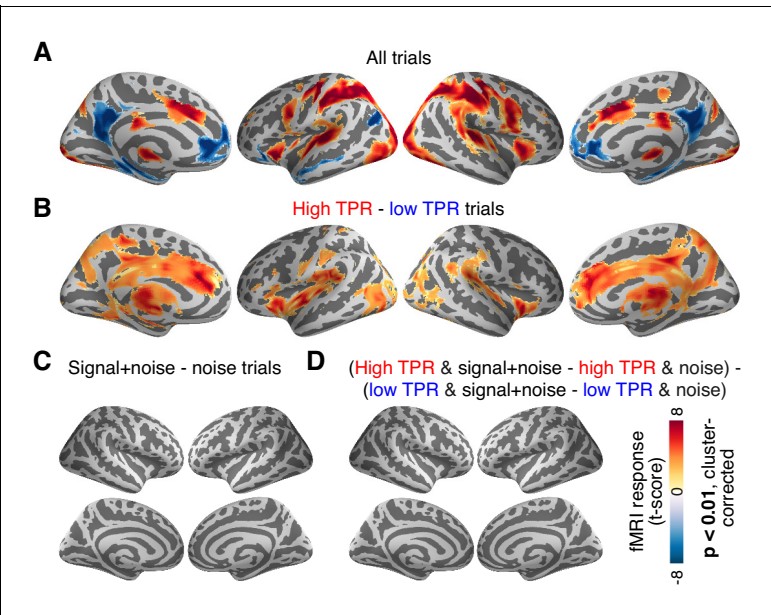

**Figure 6.** Cortex-wide fMRI correlates of phasic arousal and stimulus. (**A**) Functional map of task-evoked fMRI responses computed as the mean across all trials. (**B**) As panel A, but for the contrast high vs. low TPR trials. (**C**) As panel A, but for the contrast signal+noise vs. noise. (**D**) As panel A, but for the interaction between TPR (2 levels) and stimulus (2 levels). All panels: functional maps are expressed as t-scores computed at the group level (N = 14) and presented with cluster-corrected statistical threshold (see Materials and methods).

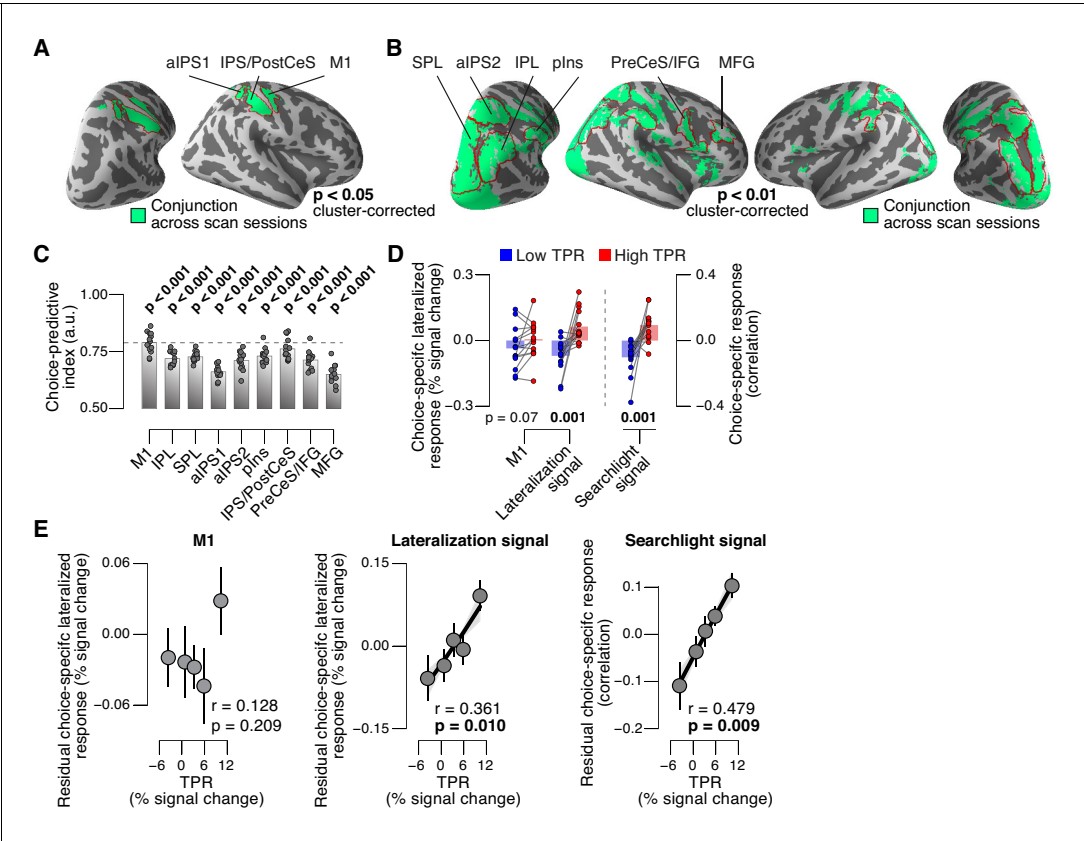

**Figure 7.** Phasic arousal predicts change of cortical decision signals. (A) Conjunction of session-wise maps of logistic regression coefficients of choice against fMRI lateralization (see *Figure 7—figure supplement 1A* for individual sessions). Tested against 0.5 at group level; red outlines, ROIs used for further analyses. (B) Conjunction of session-wise maps of searchlight choice classification precision scores (see *Figure 7—figure supplement 1C* for individual sessions). Tested against 0.5 at group level; red outlines, ROIs used for further analyses. (C) Choice-predictive indexes for choice-specific responses ('yes' vs. 'no', irrespective of stimulus; see Materials and methods and *Figure 7—figure supplement 1G*). Dashed line, index for M1, which can be regarded as a reference given the measurement noise. Data points, individual subjects. (D) Choice-specific responses, obtained through mapping lateralization (M1 and the combined 'lateralization signal', i.e., regions from *Figure 7A* excluding M1; see Materials and methods) and through searchlight classification (combined 'searchlight signal', i.e., all regions from *Figure 7B*), for low and high TPR trials. Data points, individual subjects. (E) Correlation between TPR and M1 (left), or the combined 'lateralization signal' (middle), or the combined 'searchlight signal' (right) (5 bins). In all cases, the effect of the physical stimulus was removed (see Materials and methods). Shading or error bars, s.e.m. All panels: group average (N = 14); stats, permutation test.

The following figure supplement is available for figure 7:

**Figure supplement 1.** Identifying choice-specific cortical signals.

addition to the hand area of primary motor cortex (henceforth referred to as M1), this approach yielded reliable effector-specific lateralization also in two regions of posterior parietal association cortex: the junction of the intraparietal and postcentral sulcus (IPS/PostCeS) and the anterior intraparietal sulcus (aIPS1; *Figure 7A* and *Figure 7—figure supplement 1A,B*). The second approach (*Figure 7B*) was based on multi-voxel pattern classification of choice, using a 'searchlight' procedure that scanned the entire cortex for choice information (see (*Hebart et al., 2012*, *2016*) and Materials and methods). The underlying rationale was to identify cortical regions encoding choice in other formats (e.g., in terms of more fine-grained patterns) than the hemispheric lateralization of response amplitudes. The second approach revealed robust (and reproducible) choice-specific response patterns in a number of additional regions in bilateral posterior parietal cortex and (right) prefrontal cortex: superior and inferior parietal lobule (SPL and IPL, respectively), a second region within aIPS (aIPS2), posterior insula (pIns), the junction of precentral sulcus and right inferior frontal gyrus

(PreCeS/IFG) and right medial frontal gyrus (MFG; *Figure 7B* and *Figure 7—figure supplement 1C, D*). In both approaches, choice specific regions were delineated after factoring out the physical stimulus (see Materials and methods).

In all the above choice-encoding regions, responses (estimated in a cross-validated fashion, see Materials and methods) reliably differentiated between 'yes'- and 'no'-choices – both on average (*Figure 7—figure supplement 1E,F*) and at the single-trial level (*Figure 7C*, see also *Figure 7—figure supplement 1G*). As expected, the single-trial reliability of the choice-specific responses differed between cortical regions (1-way repeated measures ANOVA with factor region of interest (9 levels): $F_{8,104} = 30.20$, $p<0.001$), with the strongest reliability for M1 (dashed horizontal line in *Figure 7C*), the region closest to the subjects' motor output.

For analysis of the association with TPR, we pooled the choice-specific signals of these different regions into three groups (*Figure 7—figure supplement 1A*): the motor end stage of the decision process M1, the combined 'lateralization signal' (i.e., regions from *Figure 7A* excluding M1), and the combined 'searchlight signal' (i.e., all regions from *Figure 7B*). Critically, as predicted, the combined choice-specific signals, but not the M1 response, were significantly pushed towards the 'yes'-choice (i.e., more positive in *Figure 7D*) for high compared to low TPR. The effect of TPR differed by cortical signal (2-way repeated measures ANOVA with factors signal type (3 levels) and TPR bin (2 levels); interaction: $F_{2,26} = 7.30$, $p=0.003$). Specifically, the difference of the choice-specific signals between low and high TPR was significantly larger for the combined lateralization signal and the combined searchlight signal than for M1 (combined lateralization signal vs. M1: $p=0.015$; combined searchlight signal vs. M1: $p=0.004$; permutation tests).

Because subjects' mean accuracy was about 74% correct, their choices were partially correlated with the physical stimulus (i.e., signal+noise vs. noise trials). Consequently, the choice-specific cortical responses were also (weakly) predictive of the stimulus (*Figure 7—figure supplement 1H*). To isolate variations in the amplitude of the choice-specific response that were independent of the stimulus, we removed (via linear regression) components explained by the stimulus and quantified the effect of TPR on the residual choice-specific cortical signals. Fitting the linear model to the combined choice-specific responses yielded highly significant TPR coefficients, for both the combined lateralization and combined searchlight signals (*Figure 7E*, middle and right panel). By contrast, the TPR-linked modulation was absent in the end stage region M1 (*Figure 7E*, left panel).

In sum, a number of fronto-parietal cortical regions exhibited signals that reliably encoded subjects' behavioral choice and were robustly modulated by phasic arousal, with a larger tendency towards the 'yes'-choice under high TPR. This was true even when factoring out the effect of the sensory evidence (i.e. presence of the target signal).

## Task-evoked pupil response are predicted by responses in a network of brainstem centers

Finally, we aimed to identify brainstem regions whose task-evoked responses were (i) linked to the trial-to-trial fluctuations of TPR, and (ii) accounted for the trial-to-trial modulation of subjects' evidence accumulation bias, and the resulting tendency to choose 'yes'. Previous work from monkey physiology has implicated three brainstem nuclei in particular in the control of TPR: the locus coeruleus (LC), the inferior colliculus (IC), and the superior colliculus (SC), respectively (*Joshi et al., 2016*; *Varazzani et al., 2015*; *Wang et al., 2012*). Here, we exploited the wide coverage of our fMRI measurements to concurrently monitor responses across a wider brainstem network, including a number of other nuclei implicated in central arousal: the dopaminergic substantia nigra (SN) and ventral tegmental area (VTA), as well as the (partly) cholinergic basal forebrain (BF). We further subdivided the BF region into the part including cell groups within the septum and the horizontal limb of the diagonal band (BF-sept) and the sublenticular part (BF-subl). BF-subl contains cholinergic neurons with widespread ascending projections (*Zaborszky et al., 2008*), which are involved in the regulation of cortical arousal state (*Lee and Dan, 2012*; *McGinley et al., 2015b*). Our analysis approach minimized the effect of physiological noise on the brainstem fMRI responses, including removal of the fourth ventricle signal (see Materials and methods). We also verified that the fourth ventricle signal was unrelated to TPR (*Figure 8—figure supplement 1D,E*). The LC region of each subject was delineated through independent structural scans (*Figure 8A*, and *Figure 8—figure supplement 1A*; for details see Materials and methods).

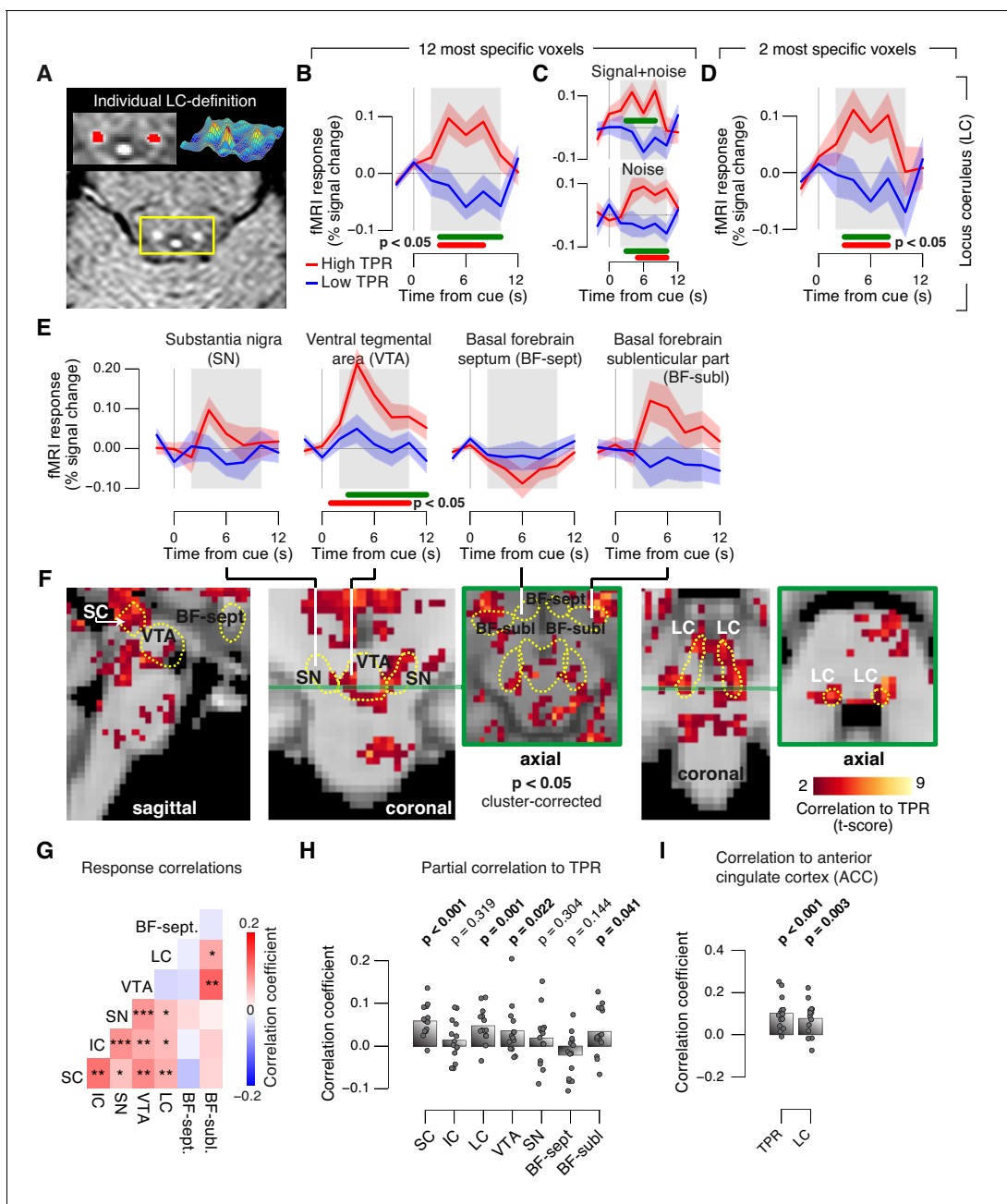

**Figure 8.** Pupil responses reflect responses of a network of brainstem nuclei. (A) Delineation of LC by structural scan. The LC corresponds to two hyper-intense spots; example subject (see *Figure 8—figure supplement 1* for all subjects). Left inset, magnification of yellow box with LC ROI. Right inset, three-dimensional representation of signal intensity levels in yellow box. (B) Task-evoked LC responses for low and high TPR. Red bar, high TPR time course significantly different from zero; green bar, high TPR time course significantly different from low TPR time course (p<0.05; cluster-corrected). Grey box, time window for computing scalar response amplitudes. (C) As panel B, but split by signal+noise and noise trials. (D) As panel B, but for the 2 voxels with highest probability of containing the LC. (E) As panel B, but for SN, VTA, and two BF-ROIs. (F) Map of single-trial correlation between TPR and evoked fMRI responses (tested against 0 at group level). Yellow outlines, brainstem nuclei from probabilistic atlases. (G) Matrix of correlations between evoked brainstem fMRI responses. Stats corrected with false discovery rate (FDR). (H) Partial correlation of evoked fMRI responses and TPR. For each ROI, responses of all other ROIs were first removed via linear regression. (I) Correlation between fMRI responses in ACC and TPR and LC. All panels: group average (N = 14); shading, s.e.m.; data points, individual subjects; stats, permutation test.

The following figure supplement is available for figure 8:

**Figure supplement 1.** TPR-linked brainstem responses.

The LC region exhibited a robust positive response on high TPR trials and a trend towards deactivation on low TPR trials (*Figure 8B–D*, and *Figure 8—figure supplement 1C*). The same pattern was evident for both signal+noise and noise trials separately (*Figure 8C*). The association to TPR was also highly significant in the most spatially specific definition of the LC region afforded by our measurements: evaluating only the two fMRI voxels with the largest probability of containing the individual LC region (*Figure 8D*, and see Materials and methods). Fluctuations of task-evoked fMRI responses measured in the LC were also robustly coupled to fluctuations in TPR amplitude at the single trial level (*Figure 8F,H*).

Similar to the LC region, we found a robust difference between low and high TPR conditions for fMRI responses in the SC and VTA regions (*Figure 8E,F*, and *Figure 8—figure supplement 1B,C*). Mapping the trial-to-trial correlations between TPR and brainstem fMRI responses at the single-voxel level yielded robust coupling to TPR in the LC, SC, VTA and as well as in BF-subl regions (*Figure 8F*).

As expected from the anatomical connectivity between brainstem centers (*España and Berridge, 2006*; *Sara, 2009*; *Wang and Munoz, 2015*), the trial-to-trial fluctuations of the task-evoked responses were significantly correlated among a number of these brainstem nuclei (*Figure 8G*). Removing components of the trial-to-trial fluctuations in TPR and fMRI responses shared with the other ROIs yielded significant residual (i.e., partial) correlations between TPR and responses in SC, LC region, VTA and BF-subl (*Figure 8H*). This indicates robust and unique contributions of these four nuclei to TPR.

Phasic brainstem responses during decision tasks might be driven by top-down signals from anterior cingulate cortex (ACC), which sends descending projections to the LC (*Aston-Jones and Cohen, 2005*) and other brainstem nuclei. In line with this notion, trial-to-trial fluctuations of both LC responses and TPR were robustly correlated to trial-to-trial fluctuations of task-evoked responses of the ACC (*Figure 8I*).

## Task-evoked responses in neuromodulatory centers, but not the colliculi, predict suppression of evidence accumulation bias

The task-evoked responses in the neuromodulatory nuclei, but not the colliculi, were tightly linked to the inferred decision computation and subjects' overt choice behavior. We computed the combined 'neuromodulatory brainstem signal' as the linear combination of responses from LC, VTA, SN, and BF that maximized the correlation to TPR (Materials and methods; correlation coefficient across subjects, 0.146 (±0.014 s.e.m.)). The amplitude of this combined signal predicted a significant reduction in conservative decision bias (*Figure 9A*), and an increased tendency to choose 'yes' (*Figure 9B*), but no change in sensitivity (*Figure 9—figure supplement 1A*). This pattern of effects was absent for the combined 'colliculi signal' (*Figure 9A,B*), a linear combination of responses from SC and IC that maximized the correlation to TPR (correlation coefficient across subjects, 0.092 (±0.011 s.e.m.)). Further, the trial-to-trial variations in the strength of the combined neuromodulatory (but not colliculi) response robustly pushed the trial-to-trial drift towards the 'yes'-boundary, in effect reducing the overall negative drift criterion (*Figure 9D*, see Materials and methods for details).

In sum, trial-to-trial fluctuations in TPR were predicted by fluctuations in the task-evoked responses of a network of brainstem regions, most notably the LC, VTA and SC. Despite the expected coupling between these and other brainstem regions (*Figure 8G*), TPR carried robust LC-, SC-, and (less strongly) VTA-specific components (*Figure 8H*). But only the responses of the neuromodulatory ROIs, not of the colliculi, accounted for the concomitant reduction of the bias in evidence accumulation and the resulting behavioral choice patterns. These results establish a tight link between phasic neuromodulator release and the dynamics of evidence accumulation.

## Discussion

Intrinsic variability in the face of uncertain evidence is a pervasive feature of decision-making (*Glimcher, 2005*; *Gold and Shadlen, 2007*; *Shadlen et al., 1996*; *Sugrue et al., 2005*; *Wyart and Koechlin, 2016*). Most current models of choice treat this intrinsic behavioral variability as a nuisance to be accounted for by additional 'noise parameters' (*Bogacz et al., 2006*; *Ratcliff and McKoon, 2008*). Other theories have proposed that the behavioral variability may be due to hidden, but systematic, biases in the decision process (*Beck et al., 2012*; *Wyart and Koechlin, 2016*). Here, we

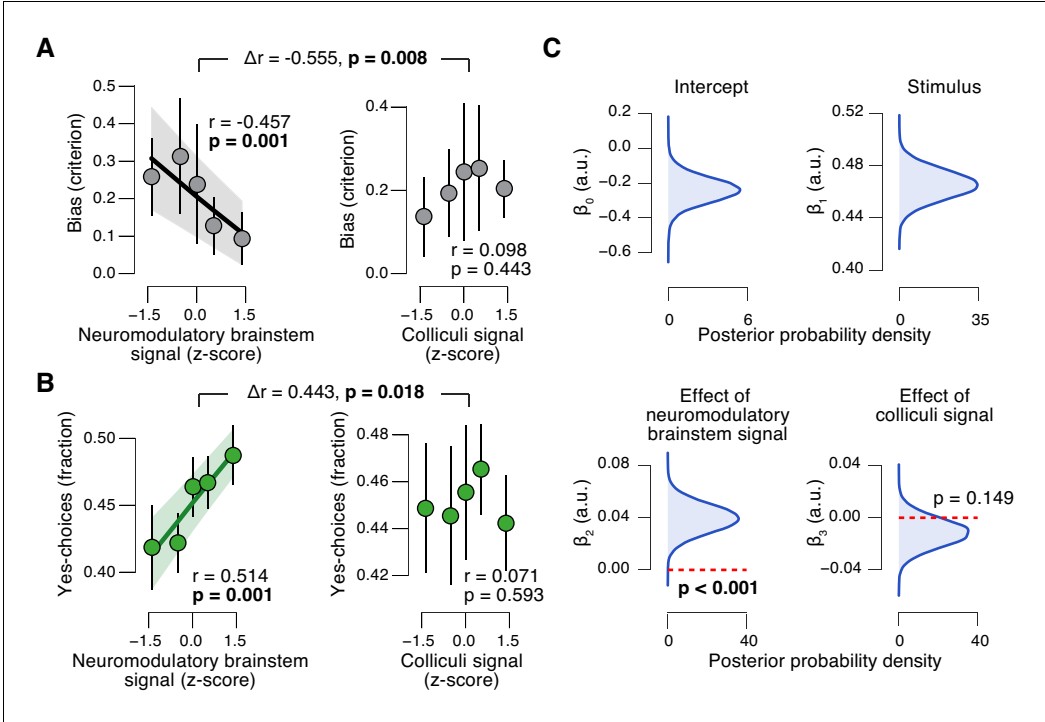

**Figure 9.** Brainstem neuromodulatory nuclei predict reduction of choice bias. (**A**) Correlation between decision bias (criterion) and the combined neuromodulatory brainstem signal (linear combination of responses in LC, SN, VTA, BF-sept, and BF-subl maximizing the correlation to TPR; see Materials and methods; left), and the combined colliculi signal (linear combination of responses in SC and IC maximizing the correlation to TPR; right) (5 bins). Stats, permutation test. (**B**) As panel A but for the correlation to fraction of 'yes'-choices. (**C**) Group-level posterior probability densities for means of parameters in the DDM regression model, through which we assessed the trial-by-trial, linear relationship between single-trial drift and the combined neuromodulatory response or the combined colliculi response (see Materials and methods; see *Figure 9—figure supplement 1* for the remaining parameters 'starting point', 'boundary separation' and 'non-decision time'). All panels: group average (N = 14); shading or error bars, s.e.m.

The following figure supplement is available for figure 9:

**Figure supplement 1.** Brainstem responses are not associated to sensitivity.

present evidence that helps reconcile these ideas. We found that a significant component of choice variability was explained by trial-to-trial variations in the amplitude of task-evoked, pupil-linked arousal responses. Specifically, pupil-linked arousal responses accounted for trial-to-trial variations in the bias of the evidence accumulation process as well as decision-related cortical population signals: under large phasic arousal conservative biases were reduced. The implication is that, without monitoring arousal responses, the associated, systematic variations in accumulation bias would appear as random trial-to-trial variability in the accumulation process (i.e., drift). Going further, we established that the dynamic bias suppression was explained by responses in a network of neuromodulatory brainstem systems controlling cortical arousal state. Taken together, our results are consistent with a scenario in which phasic neuromodulatory activity during decision-making optimizes choice behavior through a suppression of maladaptive biases in the evidence accumulation process.

## Challenges and limitations of brainstem fMRI

Imaging the brainstem with fMRI is challenging (*Astafiev et al., 2010*; *Beissner, 2015*; *Brooks et al., 2013*; *Forstmann et al., 2017*) because this region is prone to physiological noise artifacts (*Brooks et al., 2013*), and brainstem nuclei tend to be small relative to the spatial resolution of standard fMRI measurements. For example, although the adult human LC is an elongated

structure of approximately 15 mm length along the rostro-caudal axis, its diameter is only a few millimeters, as assessed by high-resolution MRI (*Figure 8A* and *Figure 8—figure supplement 1A*) (*Keren et al., 2009, 2015*). Our study addressed these challenges by following the recommendations of Eckert and colleagues (*Eckert et al., 2010*): We (i) delineated the LC in each brain, based on individual (neuromelanin-sensitive) structural MRI scans; (ii) performed fMRI tailored to the anatomical layout of the LC while maximizing functional signal-to-noise ratio (SNR), by using an in-plane spatial resolution of 2 × 2 mm and 3 mm thick slices that were oriented perpendicular to the longitudinal extent of the LC; (iii) performed no spatial smoothing of these functional data; and (iv) rigorously removed measured cardiac and respiratory signal components, as well as residual fourth ventricle signal, which have been identified as a major source of uncertainty regarding previous fMRI work on the LC (*Astafiev et al., 2010*). The resulting time-course of task-evoked fMRI responses exhibited the standard features of hemodynamic responses (*Figure 8B–E*), and correlations to pupil responses that are largely consistent with single-unit physiology in monkeys (see below). Taken together, the brainstem responses in *Figures 8* and *9* likely reflect true neural signal from brainstem nuclei, rather than physiological noise. However, there is some inevitable uncertainty regarding the spatial specificity of our measurements. Due to the lower spatial resolution of fMRI images, the co-registration between functional and structural images, and the point spread of the hemodynamic response, each fMRI voxel is likely to sample activity from brain tissue neighboring the nuclei depicted as the regions of interest (e.g., LC). Consequently, we do not conclude that the LC responses in *Figure 8B–D* reflect the activity of noradrenergic neurons only; such a conclusion would require single-unit measurements. The focus of our conclusions instead lies on the distribution of pupil and behavioral correlations across different brainstem structures, which provides an important complement to targeted single-unit measurements.

## Brainstem correlates of pupil dilation and decision bias

Despite the above-mentioned limitations, the overall distribution of pupil-linked brainstem responses shown in *Figure 8F* meaningfully follows the outlines of key candidate structures, in a fashion that is largely consistent with monkey physiology, and a previous human study on fMRI correlates of fluctuations in baseline pupil diameter (*Murphy et al., 2014a*). Our approach also identified so-far unknown effects. Previous monkey physiology has established significant coupling of pupil responses to responses of the LC, SC, and IC (*Joshi et al., 2016*; *Varazzani et al., 2015*; *Wang et al., 2012*), but not yet for dopaminergic and cholinergic structures (i.e., SN, VTA, and BF). The ability to monitor all of the above brainstem regions at once enabled quantification of their trial-to-trial correlation structure, and hence isolating the contributions that were unique to each region. This revealed that (i) many brainstem nuclei co-fluctuated during the decision task and that (ii) not only the LC and SC, but also the VTA and sublenticular part of the BF each made robust and specific contributions to task-evoked pupil dilations, over and above those shared with other brainstem centers (*Figure 8H*). Thus, the noradrenergic, cholinergic and dopaminergic systems are all phasically, and to some extent independently, recruited during challenging decision tasks, and jointly shape the concomitant changes in arousal state. Our findings provide a basis for a more comprehensive neurophysiological interpretation of results from cognitive pupillometry studies in humans.

Most importantly, we also established that only a subset of those brainstem nuclei exhibiting robust correlations with pupil responses were also predictive of the trial-by-trial suppression in decision bias. The latter effect was solely accounted for by responses in the (noradrenergic, dopaminergic, and cholinergic) neuromodulatory nuclei with diffuse projections to cortex, but not by responses in the superior or inferior colliculi. This indicates that the phasic release of neuromodulators in the brain, possibly a combination of different neuromodulators, is key for behavioral correlates of phasic arousal identified here.

## Phasic versus tonic arousal effects

A number of recent studies have characterized the relationship between tonic arousal levels (measured through baseline pupil diameter) and cortical state (*McGinley et al., 2015a*; *Reimer et al., 2014*; *Vinck et al., 2015*; *Warren et al., 2016*). Other studies have characterized the relationship between tonic arousal levels and behavioral performance (*McGinley et al., 2015a*; *Murphy et al., 2014b*). The comparison between this previous work and ours points to possible differences

between the functional correlates of tonic arousal levels and phasic, task-evoked changes in arousal. We found that phasic, task-evoked arousal responses were primarily linked to decision bias, at both, the algorithmic and cortical levels. By contrast, the above studies of tonic arousal levels have revealed effects on the quality of sensory cortical responses and behavioral sensitivity to sensory evidence (*McGinley et al., 2015b*). While we also found some evidence for non-monotonic (inverted U-shape) relationships between phasic arousal and sensitivity, the dominant and most consistent link was a monotonic and approximately linear relationship between phasic arousal and decision bias. Candidate factors accounting for these apparent differences between the functional correlates of phasic and tonic arousal might be the dynamics of the underlying neuromodulatory effects on cortical circuits, or the different combination of neuromodulatory systems involved. It will be instrumental to track TPR-linked changes in brainstem and cortical state in real time in future work.

## Post-decisional versus intra-decisional drive of phasic arousal

One account holds that the phasic arousal signals (specifically, phasic responses of the noradrenergic LC) are triggered by the bound-crossing in one of the cortical accumulator circuits; the resulting transient and cortex-wide neuromodulator release then facilitates the translation of the choice into a motor act (*Aston-Jones and Cohen, 2005*). An alternative idea (*Dayan and Yu, 2006*), supported by indirect evidence (*Cheadle et al., 2014*; *de Gee et al., 2014*), is that arousal systems are already recruited before the bound-crossing, throughout the evidence accumulation process. In line with the latter notion, we found that task-evoked pupil responses are driven most strongly by a sustained central input throughout decision formation, not only after commitment to a choice. This finding has potentially important implications for the functional role of phasic arousal in decision processing. The finding indicates that at least one of the brainstem nuclei linked to pupil responses was, likewise, activated in a sustained fashion throughout decision formation. The resulting neuromodulatory transients might alter the state of brain regions involved in decision computations as the decision unfolds, provided that the accumulation operates on timescales of seconds or longer. Because the tasks used in previous animal physiology studies of task-related LC responses involved much faster decision processes than the one studied here (reaction times of about 0.5 s vs. 2 s, respectively), it remains unknown whether the more sustained, task-evoked responses also occur in noradrenergic neurons (but see [*Varazzani et al., 2015*]). Sustained responses encoding reward uncertainty have been observed in dopaminergic neurons in the VTA (*Fiorillo et al., 2003*), one of the structures whose task-evoked responses predicted pupil responses. Future electrophysiological studies should determine the time course of task-related activation in the different nuclei of the brain's arousal network during sensory-motor decisions involving protracted evidence accumulation (*Nomoto et al., 2010*).

## How do diffuse neuromodulatory signals translate into specific effects on choice behavior?

One notable aspect of our findings is that the functional correlates of neuromodulatory responses were specific for a particular choice option (see *Figure 9*). Also in the context of learning, the interplay between pupil-linked arousal and competitive cortical circuitry has been found to translate into specific effects on cognition and behavior (*Eldar et al., 2013*).

A scenario consistent with our results is that phasic neuromodulator release alters the relative strength of information flow between cortical processing stages, suppressing 'top-down' relative to 'bottom-up' signals (*Friston, 2010*; *Gil et al., 1997*; *Hsieh et al., 2000*; *Kimura et al., 1999*; *Kobayashi et al., 2000*). In perceptual decisions like the ones studied here, early sensory cortices provide bottom-up sensory likelihood signals, while top-down signals might encode prior beliefs (*Friston, 2010*; *Pouget et al., 2013*). Thus, through a relative suppression of 'top-down' signal flow, phasic arousal might reduce the weight of the prior (reflecting subjects' intrinsic bias) relative to the likelihood. Specifically, in our yes-no task, the prior may have been a conservative bias for choosing 'no'. Reducing its weight would reduce this bias. Such an increase in the relative weight of bottom-up signals might be implemented by synaptic gain modulation through neuromodulators. This gain modulation, in turn, might depend on the precision (inverse of uncertainty) of incoming sensory data (*Friston, 2010*; *Moran et al., 2013*).

The above scenarios postulate a uni-directional effect of neuromodulatory transients on cortical decision computations. However, this interaction may also be bi-directional, with trial-to-trial fluctuations of cortical decision signals driving fluctuations of phasic arousal responses (*Aston-Jones and Cohen, 2005*; *Dayan and Yu, 2006*). Specifically, phasic LC responses may be driven by specific cortical regions (e.g., the ACC), which compute the ratio of the posterior probability of target presence over the (estimated) prior probability of target occurrence (*Dayan and Yu, 2006*). The resulting phasic norepinephrine release across cortex might reset cortical networks (*Bouret and Sara, 2005*) and interrupt the (default) state encoding the prior (*Dayan and Yu, 2006*). In a yes-no task such as ours, a tendency towards the 'no'-option may correspond to the default state for conservative subjects, and a phasic arousal signal is generated when decision-related neural activity ramps towards 'yes', facilitating the transition of the entire cortical system towards that non-default state.

## Conclusion

Our findings establish that phasic task-evoked pupil responses during the formation of sensory-motor decisions reflect responses of a network of neuromodulatory brainstem centers including the noradrenergic LC. Phasic, pupil-linked arousal alters choice-encoding population signals in parietal and prefrontal association cortices. Phasic arousal in general, and neuromodulatory brainstem responses in particular, explain a dynamic reduction in decision-makers' bias towards one particular choice. The resulting trial-to-trial variability of decision bias accounts for a significant component of the intrinsic behavioral variability: when decisions are made in the face of uncertainty, tracking phasic arousal signals may be just as important for predicting choice behavior as tracking the objective evidence gathered from the outside world.

# Materials and methods

## Subjects

We report analyses of four independent data sets, from behavioral tasks described in the subsequent section. All subjects had normal or corrected-to-normal vision and gave written informed consent. Subjects received €15 per hour (all visual tasks) or research credit (auditory task) for their participation. The ethics committee of the Psychology Department of the University of Amsterdam approved the experiments.

Fifteen healthy subjects (5 females; age range, 22–35 y) participated in the main experiment of this study, entailing concurrent pupillometry and brainstem as well as cortical fMRI recordings. Here, each subject participated in several fMRI sessions: one to define retinotopically organized visual cortical areas (75 min) and two sessions (three for one subject) for the main experiment (about 2 hr per session). Three subjects were authors, and the remaining 12 subjects were naive to the purpose of the study. The results were unchanged when excluding the three authors (see *Author response*, online) and the one subject who performed three sessions (and more trials; see section *Behavioral tasks*) of the main experiment. One (male) subject was excluded from the analyses because the stimulus software did not receive the triggers from the MRI scanner in two sessions (the age range remained the same).

We also re-analyzed the 21 subjects from an existing behavioral data set, for which we had previously published different analyses (*de Gee et al., 2014*) (*Figure 2*, *Figure 2—figure supplement 1*, *Figure 4* and *Figure 4—figure supplement 1*). In that experiment, 23 subjects had performed a yes-no visual contrast detection task with trial structure analogous to that of the fMRI experiment, enabling joint fitting of the drift diffusion model to both data sets using a hierarchical Bayesian procedure (see below). To this end, we excluded the two subjects from the (*de Gee et al., 2014*) data set who had also participated in the current fMRI experiment, keeping the two samples independent.

Finally, 24 subjects (20 females; age range, 19–23 y) performed an auditory tone-in-noise detection task (*Figure 3A,B*), and 15 subjects (six females; age range, 23–37 y) performed a visual random dot motion discrimination task (*Figure 3C,D*).

## Sample sizes

The sample sizes were determined based on a number of criteria: (i) the assessment of the behavioral correlates of TPR obtained in a previous study (*de Gee et al., 2014*); (ii) the need to obtain as many trials as possible from each individual (necessary for detailed modeling of choice behavior as a function of TPR within subjects at a first level, before second-level statistics; *Figures 2–4*); and, in the case of fMRI, (iii) the need to obtain detailed retinotopic maps per individual from a separate scanning session (*Figure 5*), as well as robust maps of choice-specific activity by means of conjunction across two sessions of the main experiment (*Figure 7*, and *Figure 7—figure supplement 1*). Taken together, these criteria prioritized obtaining a large amount of data (and experimental sessions) from each participant, which was traded off against the total number of participants.

## Behavioral tasks

### Main task: Visual contrast detection (yes-no)

Each trial began with the central fixation dot turning green and consisted of three consecutive intervals (*Figure 1A*): (i) the baseline interval (2 s; containing only noise); (ii) the decision interval, the start of which was signaled by the occurrence of a tone (200 ms duration) and which was terminated by the subject's response (or after a maximum duration of 3.5 s); (iii) the inter-trial interval (ITI), which consisted of a dark grey fixation dot on an otherwise blank screen and was uniformly distributed between 4 and 12 s.

A dynamic noise pattern (refresh rate: 60 Hz) was presented throughout the trial. The luminance across all pixels was kept constant. This pedestal noise pattern had 10% contrast and was refreshed on each frame. On one half of trials ('signal+noise' trials), a sinusoidal grating (five cycles per degree) was superimposed on the visual noise for the entire decision interval, from the onset of the auditory cue to the subject's motor response (*Figure 1A*). The other half of trials ('noise' trials) contained no target signal during the decision interval. Signal presence was randomly selected on each trial, under the constraint that it would occur on 50% of the trials within each block of 40 trials. All stimuli were presented in a Gaussian annulus, with an average distance (±SD) to fixation of 1.8 (0.6) degrees (*Figure 1A*). In different blocks, the target grating, if present, was tilted 45° (clockwise, CW) or 135° (counter-clockwise, CCW). To minimize uncertainty, subjects were informed about the orientation of the target before each block, by means of a full-contrast presentation of the target signal.

Subjects were instructed to report the presence or absence of the signal by pressing one of two response buttons with their left or right index finger, once they felt sufficiently certain (free response paradigm). The mapping between perceptual choice and button press (e.g., 'yes' –> press right key; 'no' –> press left key) was counterbalanced across subjects. At the end of each block of 40 trials subjects were informed about their performance.

Throughout the main experiment, the contrast of the target signal was fixed at a level that yielded about 75% correct choices. Each subject's individual threshold contrast was determined before the main experiment in the MRI scanner (during anatomical scans), using an adaptive staircase procedure (Quest). Here, we used a two-interval forced choice variant of the contrast detection task (one interval: signal+noise, the other: noise). The corresponding threshold contrasts yielded a mean accuracy of 73.79% correct (±1.32 % s.e.m.) in the yes-no visual contrast detection task during fMRI.

Subjects performed between 10 and 12 blocks (distributed over two scanning sessions), yielding a total of 400–480 trials per subject. One subject performed a total of 16 blocks (distributed over three scanning sessions), yielding a total of 640 trials.

Stimuli were back-projected on a transparent screen using a gamma-corrected LCD projector with a spatial resolution of 1920 × 1200 pixels, run at a vertical refresh rate of 60 Hz. Subjects were supine in the MRI scanner and viewed the screen from 120 cm via a mirror attached to the head coil. To minimize any effect of light on pupil diameter, the overall luminance of the screen was held constant throughout the experiment.

### Auditory tone detection task (yes-no)

The trial structure was identical to that of the visual contrast detection task described above, but without noise during the pre-decision interval baseline, and with shorter ITIs (uniformly distributed between 1 and 2 s). The decision interval consisted of an auditory noise stimulus (as in

(*McGinley et al., 2015a*), or a pure sine wave (2 KHz) superimposed onto the noise (50% of trials each). These stimuli were presented from the onset of the auditory stimulus to the subject's motor response. Threshold volumes (in dB), determined beforehand via an adaptive staircase procedure, yielded a mean accuracy of 73.41% correct (±0.90 % s.e.m.). Auditory stimuli were presented using an IMG Stageline MD-5000DR over-ear headphone, suppressing ambient noise. Subjects performed between 11 and 13 blocks in the behavioral lab (distributed over two measurement sessions; same set-up as in [*de Gee et al., 2014*]), yielding a total of 1320–1560 trials per subject.

## Visual motion discrimination task (two-alternative forced choice)

The trial structure was identical to that of the visual contrast detection task described above, but with fixed stimulus duration (750 ms; interrogation protocol) and visual feedback at the end of each trial (green/red rectangle at fixation to signal correct/error). An auditory white noise stimulus (250 ms) was played on 50% of the trials. The mapping between perceptual choice and button press (e. g., 'up' –> press right key; 'down' –> press left key) varied from session to session, counterbalanced within subjects.

Random dot motion stimuli (refresh rate: 120 Hz) were presented throughout the experiment in a central annulus (outer diameter 16.8°, inner diameter of 2.4°) around the central fixation rectangle (0.45° length). The annulus contained 524 dots all within one hemifield (half circle). Hemifield presentation (left or right) changed across blocks and was counterbalanced across subjects and sessions. Dots were 0.15° in diameter and white, presented on a grey background. Dots were divided into 'signal' and 'noise' dots, the proportion of which defined the motion coherence level. Outside the stimulus interval, motion coherence was fixed to zero (pure noise). During the stimulus interval, signal dots moved at 7.5°/s in one of two directions (up or down), were randomly selected on each frame, had a lifetime of 10 frames, and were re-plotted in random locations thereafter (reappearing on the other side when their motion extended outside of the annulus). Noise dots were randomly assigned to locations within the annulus on each frame. Independent motion sequences (n = 3) were interleaved to prevent tracking of individual dots (*Pilly and Seitz, 2009*). Threshold motion coherence, determined beforehand via an adaptive staircase procedure, levels yielded a mean accuracy of 88.86% correct (±1.49 % s.e.m.) and 71.01% correct (±1.35 % s.e.m.), respectively. We collapsed across these two peri-threshold levels, to maximize statistical power, but we verified that the effect found was analogous for both difficulty levels.

Subjects performed between 23 and 24 blocks (distributed over four sessions; same set-up as in the main task, except that stimuli were presented on a 31.55' MRI compatible LCD display with a spatial resolution of 1920 × 1080 pixels) yielding a total of 575–600 trials per subject. One subject performed a total of 18 blocks (distributed over three scanning sessions), yielding a total of 450 trials. Here, we analyzed only the 50% of trials without the auditory white noise stimulus (see above), which are comparable to the trials from the other tasks. Data from one measurement session of one subject was excluded from the analyses because of poor eye-tracker data quality.

## Magnetic resonance imaging data acquisition

For the main experiment, MRI data were acquired on a 3T Philips Achieva XT MRI scanner using a 32-channel head coil in two types of sessions: retinotopic mapping sessions (for defining the borders of visual cortical areas V1-V3, see section *Definition of regions of interest*) and main experimental sessions. In all sessions, cardiac cycle was monitored with a pulse oximeter attached to the left index finger, and respiratory activity was recorded with a chest belt, for physiological noise removal. Both physiological signals were recorded at a sampling rate of 496 Hz.

During the main experimental sessions, EPI images were acquired in 35 slices (thickness: 3.0 mm, no gaps) oriented perpendicular to the floor of the fourth ventricle (i.e., perpendicular to the longitudinal extent of the locus coeruleus (*Keren et al., 2009*), with the following parameters: TR = 2 s, TE = 27.62 ms, flip angle = 76.1°, SENSE acceleration factor = 3.0. Images were acquired at an in-plane resolution of 2.0 × 2.06 mm and were reconstructed at a resolution of 1.79 × 1.79 mm. A structural T1 scan was acquired with an MPRAGE sequence for anatomical co-registration and cortical surface reconstruction (voxel size: 1 × 1 × 1 mm, TR = 8.2 ms, TE = 3.73 ms, flip angle = 8°). An additional structural scan was acquired with a T2-weighted sequence and higher resolution than the EPI scans (1 × 1 × 1.5 mm, TR = 5114 ms, TE = 12.5 ms, flip angle = 90°) to facilitate co-registration

of EPI images and the high-resolution structural T1 scan. Two turbo spin echo (TSE) neuromelanin-sensitive structural scans were acquired for delineation of the LC (*Keren et al., 2015*, *2009*; *Shibata et al., 2007*), again oriented perpendicular to the floor of the fourth ventricle. The first (partial field-of-view) TSE scan was obtained with the following parameters: 20 slices (1.5 mm, no gaps), in-plane resolution: $0.7 \times 0.88$ (reconstructed at: $0.35 \times 0.35$ mm), TR = 500 ms, TE = 10 ms, flip angle = 90°, covering the brainstem only. The second (whole-brain) TSE scan was obtained with the following parameters: 35 slices (3 mm, no gaps), in-plane resolution: 1.96 x 2.0 (reconstructed at: $0.47 \times 0.47$ mm), TR = 500 ms, TE = 10 ms, flip angle = 90°. Finally, field maps were acquired using two separate acquisitions (voxel size: $2 \times 2 \times 2$ mm$^3$, TR = 11 ms, TE$_1$ = 3.0, TE$_2$ = 3.5, ms, flip angle = 8°)

During retinotopy sessions, EPI scans were acquired in 29 slices (thickness: 2.5 mm, with 0.25 mm slice gaps) with the following parameters: in-plane resolution: $2.5 \times 2.58$ mm (reconstructed at $2.5 \times 2.5$ mm), TR = 1.5 s, TE = 27.62 ms, flip angle = 70°, SENSE acceleration factor = 3.0. An additional structural scan was acquired with a T2-weighted sequence and higher resolution than the EPI scans ($1.25 \times 1.25 \times 1.25$ mm with 0.12 mm slice gaps, TR = 8390 ms, TE = 100 ms, flip angle = 90°) to facilitate co-registration of EPI images and the high-resolution structural T1 scan used for cortical surface reconstruction (see above).

## Eye data acquisition

Concurrently with the fMRI recordings, the left eye's pupil was tracked (via the mirror attached to the head coil) at 1000 Hz with an average spatial resolution of 15 to 30 min arc, using an EyeLink 1000 Long Range Mount (SR Research, Osgoode, Ontario, Canada). The MRI-compatible (non-ferro-magnetic) eye tracker was placed outside the scanner bore, and it was calibrated once at the start of each scanning session. The purely behavioral experiments were conducted in a psychophysics laboratory. Here, the left eye's pupil was also tracked at 1000 Hz with an average spatial resolution of 15 to 30 min arc, using the same EyeLink 1000 system (SR Research, Osgoode, Ontario, Canada).

## Analysis of task-evoked pupil responses

### Preprocessing

Periods of blinks and saccades were detected using the manufacturer's standard algorithms with default settings. The remaining data analyses were performed using custom-made Python software (*de Gee, 2017a*, https://github.com/jwdegee/2017_eLife (with a copy archived at https://github.com/elifesciences-publications/2017_eLife); *de Gee, 2017b*, https://github.com/jwdegee/2014_PNAS). We applied to each pupil recording (i) linear interpolation of values measured just before and after each identified blink (interpolation time window, from 150 ms before until 150 ms after blink), (ii) band-pass filtering (third-order Butterworth, passband: 0.01–6 Hz), (iii) removal of pupil responses to blinks and to saccades, by first estimating these responses by means of deconvolution and then removing them from the pupil time series by means of multiple linear regression (*Knapen et al., 2016*), and (iv) conversion to units of modulation (percent signal change) around the mean of the pupil time series from each block. Filtering (specifically the lower cutoff at 0.01 Hz) was performed on both fMRI and pupil time series, ensuring equal treatment of both signals to be correlated.

### Quantification of task-evoked pupillary responses (TPR)

We computed task-evoked pupillary response (TPR) amplitude measures for each trial as the mean of the pupil diameter modulation values in the window $-1$ s to 1.5 s from choice (same time window as in *de Gee et al., 2014*]), minus the mean baseline pupil value during the 0.5 s before the cue (i.e., decision interval onset) (*Figure 1B*). The sluggish hemodynamic system and, to a lesser extent, the peripheral pupil apparatus (*Hoeks and Levelt, 1993*; *Korn and Bach, 2016*) act as temporal low-pass filters. As a result, trial-to-trial variations in reaction time (RT; and thus, the duration of the task-related sustained activity, see *Figure 1D*, and *Figure 1—figure supplement 1A–C*) can induce trial-to-trial variations of the fMRI and TPR amplitudes, without changes in the amplitude of the underlying neural responses. Indeed, there was a robust relationship between reaction time and TPR (*Figure 1—figure supplement 1D*). Therefore, to specifically isolate trial-to-trial variations of underlying response amplitudes, variations due to RT were removed via linear regression from both the TPR

and fMRI responses. This was done for all analyses reported in the main text. Hereby, RT was defined as the time from decision onset (cued by tone) until the button press. To establish the robustness of the effects reported here we verified that all pupil-linked behavioral results were evident also without removing RT-related components (*Figure 2—figure supplement 1*, and *Figure 4—figure supplement 2*).

In the majority of analyses, trials were sorted by TPR amplitude and collapsed into three bins containing the lowest and highest 40% (which were used for analyses), as well as the intermediate 20% of TPR amplitudes (*Figure 1B,C*). This achieved a trade-off between maximizing both (i) trial counts in the high and low TPR bins and (ii) the disparity between the TPR amplitudes for both bins. In other analyses, we used five equally populated bins of single-trial TPR amplitudes (*Figure 1—figure supplement 1*, *Figure 2*, *Figure 2—figure supplement 1*, *Figure 3* and *Figure 7*). In *Figure 3D*, we used four bins because of the lower trial count per subject in this data set after excluding the trials with the auditory white noise manipulation (see section *Behavioral tasks*)

## General linear modeling of TPR

We used a general linear modeling approach described in detail in a previous paper (*de Gee et al., 2014*) to estimate the relative contribution of three different putative temporal input components to the peripheral apparatus controlling pupil motility (*McDougal and Gamlin, 2008*). We (i) cut the pupil time series into single-trial epochs ranging from 1 s before cue to 5 s after cue, (ii) baseline-corrected each epoch by subtracting the mean baseline pupil value during the 0.5 s before cue, and (iii) concatenated these baseline-corrected epochs into a new time series excluding large parts of the inter-trial intervals. The GLM was then fit to this new, cleaned-up time series. The GLM consisted of the following transient events (*Figure 1D*, and *Figure 1—figure supplement 1A–C*): cue (onset of decision interval) and the choice. The transient corresponding to choice was placed at 0.24 s before button press, adopted from a report quantifying the interval between phasic LC activity and behavioral response in a forced-choice task in monkey (*Clayton et al., 2004*). The GLM also consisted of a sustained component in between the two transient events, which was modeled as a boxcar function. We normalized the boxcar regressor by dividing the height of the boxcar by the number of samples in that particular interval, such that this regressor had the same norm as the transient regressors. Thus, estimated beta weights were comparable between both sets of regressors. Each regressor was then convolved with a canonical pupil impulse response function (parameters taken from [*Hoeks and Levelt, 1993*]), and multiple regression yielded the best-fitting beta weights for each regressor type (i.e., temporal component of the pupil response), separately for each subject.

## Analysis and modeling of choice behavior

The first trial from each block and trials in which subjects failed to respond within the time limit of 3.5 s (see section *Stimuli, task and procedure*) were excluded from all analyses. RT was defined as the time from decision interval onset (cued by tone) until the button press. In a model-free analysis, we computed the fraction of 'yes'-choices separately for two TPR bins (*Figure 2*, *Figure 2—figure supplement 1*, and *Figure 3*). To ensure that each TPR bin consisted of the same number of signal+noise and noise trials, we (i) sorted all trials of a subject into of four 'cells' defined by the factors TPR (low and high) and stimulus (signal+noise and noise), (ii) determined the lowest trial count across the four cells, (iii) randomly sampled the same number of trials (without replacement) from the remaining cells, and (iv) computed the fraction of 'yes'-choices separately for the two TPR bins. We then repeated this procedure 1000 times and averaged the results across all repetitions. The fraction of non-preferred choices (*Figure 3C*) was computed in the same way, with the exception that the non-preferred choice was defined as the choice opposite to the subject's overall bias (towards up or down) calculated across all trials (i.e., irrespective of TPR). We then modeled the effects of phasic arousal (as indexed by TPR) on choice behavior using two approaches, which yielded converging results.

## Signal-detection theoretic (SDT) modeling

In a first approach, we computed the SDT-metrics d' and criterion (*Green and Swets, 1966*) separately for multiple (two or five) bins of TPR (*Figure 2*, *Figure 2—figure supplement 1*, and *Figure 3*) or combined brainstem response (*Figure 9*, and *Figure 9—figure supplement 1*). We estimated d'

as the difference between z-scores of hit- and false-alarm rates. We estimated criterion by averaging the z-scores of hit- and false-alarm rates and multiplying the result by $-1$.

We used sequential polynomial regression analysis (*Draper and Smith, 1998*) to quantify the dependence of d' and criterion on TPR. This procedure allowed us to systematically test whether TPR predominantly exhibited no (zero-order polynomial), a monotonic (first-order polynomial), or a non-monotonic (second-order polynomial) effect on the SDT metrics. The SDT metric *y* was modeled as a linear combination of polynomial basis functions of 5 TPR bins:

$$y \sim \beta_0 + \beta_1 TPR^1 + \beta_2 TPR^2 \tag{1}$$

with $\beta$ as polynomial coefficients. The corresponding regressors were orthogonalized, and each model was sequentially tested in a serial hierarchical analysis, based on *F*-statistics. This analysis was performed at the group level, and it tested whether adding the next higher order model yielded a significantly better description of the response than the respective lower order model. We tested models from the zero-order (constant, no effect of TPR) up to the second-order (quadratic, non-monotonic). If the first-order model was significantly better than the zero-order model at the group level, we fitted a linear model and tested the corresponding linear correlation coefficients across the group. This was true for SDT criterion in all cases, except *Figure 3D*, in which the linear expansion was only marginally significant; however, the linear correlation was highly significant when tested across the group. If the second-order model was significantly better than the first-order model at the group level, we fitted a quadratic model between TPR and behavior for each subject and tested the second-order coefficients across the group. This was true for SDT d' in some of the cases (*Figures 2E* and *3B*).

Having established a robust first-order (monotonic) relationship between TPR and SDT criterion, we then characterized the timing of this effect by means of a sliding window (linear) correlation analysis over the interval from 1 s before cue to 3 s after response (window length: 250 ms, step size: 25 ms). We computed separate, baseline-corrected TPR values (see section *Quantification of task-evoked pupillary responses*) for each position of the window. Per time window, we then sorted trials by the TPR-values into five bins, and correlated these values with criterion estimates for the corresponding bins. This yielded time courses of the correlation between TPR and criterion.

## Drift diffusion modeling

In the second approach, we fitted the drift diffusion model (*Ratcliff and McKoon, 2008*) to RT distributions for 'yes'- and 'no'-choices, separately for low and high TPR trials (*Figure 4*, *Figure 4—figure supplement 2*). We fitted the model using the hierarchical Bayesian implementation of the HDDM toolbox (*Wiecki et al., 2013*) (version 0.6). The group distribution constrains individual subject parameter estimates, with a stronger influence when its variance is estimated to be small (for details of the procedure, see [*Wiecki et al., 2013*]). Fitting the model to RT distributions for 'yes'- and 'no'-choices (termed 'stimulus coding' in [*Wiecki et al., 2013*]), as opposed to the more common fits of correct and incorrect choice RTs (termed 'accuracy coding' in [*Wiecki et al., 2013*]), was essential for estimating parameters that could have induced biases in subjects' behavior.

We fit the model to the behavioral data from a total of 35 subjects: 14 subjects from the current fMRI study, and 21 subjects from a previous study employing an analogous contrast detection task ([*de Gee et al., 2014*]; see section *Subjects*). Doing so improved the estimation of the group-level distribution over parameters, which was used to constrain the individual subject parameter estimates. Specifically, the pooling was required to jointly fit the parameters starting point and drift criterion, both of which can account for changes in decision bias, but via different mechanisms, and are distinguishable through their distinct effects on the shape of the RT distribution (*Figure 4—figure supplement 1*). In our fits, we allowed the following parameters to vary between low and high TPR: (i) the mean drift rate across trials; (ii) the drift criterion (an evidence-independent constant added to the drift toward one or the other bound); (iii) the separation between both decision bounds (i.e., response caution); (iv) the starting point of the accumulation process; (v) the non-decision time (sum of the latencies for sensory encoding and motor execution of the choice). Drift rate variability is an additional parameter that was found to improve fits to empirical RT distributions (*Ratcliff and McKoon, 2008*). However, this parameter is prone to fit error (*Ratcliff and Childers, 2015*) and we

had no a priori hypothesis about its relationship to phasic arousal. We therefore fit drift rate variability to all data (i.e., regardless of TPR), to maximize robustness of our fits.

To test the robustness of the significance of the TPR-dependent effect on drift criterion, we re-fitted the model, but now fixing drift criterion with TPR, while still allowing all other of the above parameters to vary with TPR. Using deviance information criterion (*Spiegelhalter et al., 2002*) for model selection, we compared whether the added complexity of our original model was justified to account for the data. This is a common metric for comparing hierarchical models, for which a unique 'likelihood' is not defined, and the effective number of degrees of freedom is often unclear (*Spiegelhalter et al., 2002*).

Finally, we used the HDDM toolbox (*Wiecki et al., 2013*) to assess the trial-by-trial, linear relationship between the combined neuromodulatory brainstem responses, the combined colliculi responses, and the drift (*Figure 9C*, *Figure 9—figure supplement 1*). We fitted a variant of the above-described model (*Figure 4A*) to RT distributions for 'yes'- and 'no'-choices from the 14 subjects from the current fMRI study, but now modeling the drift on each trial as the following linear combination:

$$v \sim \beta_0 + \beta_1 S + \beta_2 M + \beta_3 C \tag{2}$$

where $v$ was the single-trial drift, $S$ was a binary vector describing the stimulus identity (1, signal +noise; $-1$, noise), $M$ was a vector of the single-trial combined neuromodulatory brainstem response (see section *Quantification of task-evoked fMRI responses and correlation with TPR*), $C$ was a vector of the single-trial combined colliculi response. The fit parameters quantified how the drift on single trials was affected by the overall drift bias (i.e., mean drift criterion across trials, regardless of brainstem response, $\beta_0$), the overall drift rate (i.e., mean stimulus-dependent drift across trials, $\beta_1$), and the neural responses of the combined neuromodulatory nuclei or colliculi ($\beta_2$ and $\beta_3$), respectively. The parameters starting point, boundary separation, and non-decision time were also included in the model, but not as a function of either of the combined brainstem responses.

## Analysis of MRI data

MRI data were analyzed using custom-made software written in Python (*de Gee, 2017a*, https://github.com/jwdegee/2017_eLife; a copy is archived at https://github.com/elifesciences-publications/2017_eLife). A number of processing steps relied on FSL (RRID:SCR_002823, *Smith et al., 2004*) and FreeSurfer (RRID:SCR_001847, *Dale et al., 1999*; *Fischl et al., 1999*).

### Preprocessing

T1-weighted anatomical scans acquired at the beginning of each scanning session were automatically segmented and inflated for visualization using FreeSurfer (*Dale et al., 1999*; *Fischl et al., 1999*). We then applied to the EPI scans (i) removal of non-brain tissue (brain extraction) using the BET tool in FSL, (ii) unwarping using a B0 field map and FUGUE (FMRIB's Utility for Geometrically Unwarping EPI's), (iii) image realignment to compensate for small head movements (*Jenkinson et al., 2002*) (for improved precision we used as the target volume the high-resolution T2-weighted anatomical scan; the resulting up-sampled EPI scans were down-sampled back to their original resolution), (iv) high-pass filtering to correct for baseline drifts in the signal (Gaussian-weighted least-squares straight line fitting, with window size = 50 samples), and (v) conversion to units of modulation (percent signal change) around the mean fMRI series. We concatenated all EPI volumes preprocessed in that way across blocks within one scanning session. We applied physiological noise correction using FSL PNM (*Brooks et al., 2013*), an extended version of RETROICOR (*Glover et al., 2000*), whereby cardiac and respiratory phases were assigned, separately for each slice, to each volume in the concatenated EPI image time series.

Our complete physiological noise regression model included 34 physiological noise regressors (*Brooks et al., 2013*): 4th order harmonics to capture the cardiac cycle, 4th order harmonics to capture the respiratory cycle, 2nd order harmonics to capture the interaction between the cardiac and respiratory cycles, 2nd order harmonics to capture the interaction between the respiratory and cardiac cycles, one regressor to capture heart rate, and one regressor to capture respiration volume. These 34 noise predictors, plus two for fMRI responses to eye-blinks and saccades (obtained by convolving blink and saccade events with a canonical hemodynamic impulse response function),

were regressed against the time series of EPI volumes using multiple linear regression. The residual (i.e., noise-corrected) time series were used for all further analyses.

Subsequent analyses proceeded along two separate pipelines: (i) in functional space, by extracting responses from several, specifically delineated regions of interest (ROIs); or (ii) in anatomical space, as voxel-wise functional maps. The procedures for the delineation of ROIs are described in the section *Definition of regions of interest (ROIs)* below. For the computation of group average TPR-brainstem correlation map (*Figure 8F*) and of the cortex-wide functional maps (*Figure 6*, *Figure 7A,B*, and *Figure 7—figure supplement 1A–D*), the time series of EPI volumes were first transformed to MNI space (affine transformation with 12 degrees of freedom and sinc interpolation with FSL FLIRT). For the computation of cortex-wide functional maps, we additionally applied spatial smoothing to all EPI volumes in MNI space using isotropic Gaussian filter kernels; the full width at half maximum (FWHM) of the kernels was 8 mm for all maps except for those of searchlight precision scores (*Figure 7B*, and *Figure 7—figure supplement 1C,D*). All other analyses reported in this paper were performed without spatial smoothing.

## Quantification of task-evoked fMRI responses and correlation with TPR

The slow event-related design (mean ITI: 8 s) enabled us to quantify task-evoked fMRI responses for each trial as the difference between fMRI measurements during the trial (starting from 2 s from cue) and the mean fMRI response during the pre-decision baseline interval ($-2$ s to 2 s from cue). Scalar fMRI response amplitudes were computed by collapsing response values across the interval 2 s to 12 s from cue. Trial-to-trial variations of each voxel's task-evoked fMRI response amplitude due to RT were removed (via linear regression), to isolate fMRI response variations that were due to variations in the amplitude of the underlying neural responses (see section *Analysis of task-evoked pupil responses*). For the analysis of evoked fMRI responses in the brainstem (*Figure 8*, and *Figure 8—figure supplement 1*), we additionally removed (via linear regression) signal fluctuations from the fourth ventricle (delineated based on the TSE scan by averaging across all voxels covering the ventricle) from the time series from each brainstem ROI or voxel, before computing task-evoked responses.

We computed maps of task-evoked fMRI responses as the voxel-wise difference between fMRI response during the decision interval and during baseline. The resulting maps were computed separately for each subject and then tested against 0 across the group (see section *Statistical comparisons*). We computed the following statistical maps: (i) overall task-evoked response (*Figure 6A*), (ii) difference between high and low TPR trials (*Figure 6B*), (iii) difference between signal+noise and noise trials (*Figure 6C*), and (iv) interaction between TPR and stimulus (*Figure 6D*), whereby the interaction map (iv) was computed as the difference between the two difference maps (ii) and (iii).

We verified that the results from *Figure 8* and *Figure 8—figure supplement 1* were not affected by baseline fluctuations caused by variations in ITI (the analyses in *Figure 5* and *Figure 7* were based on stimulus or choice-specific components of the fMRI response, which are unlikely to be affected by non-specific baseline response fluctuations). In a control analysis, we repeated the brainstem response estimates from *Figure 8* after excluding 50% of all trials with the shortest ITIs. We also verified that the results were not due to non-linearity (i.e., floor and ceiling effects) in the TPR measurements. To this end, we repeated the analyses from *Figure 8* after removing the 20% of trials with extreme overall pupil size during the trial (top and bottom 10% pupil measurements in the TPR-window, without subtracting the pre-trial baseline pupil). Both sets of control analyses yielded qualitatively identical results as in *Figure 8*, and *Figure 8—figure supplement 1* (data not shown).

We computed the correlation between TPR and task-evoked fMRI responses across all trials (*Figure 8F*) after first removing (via linear regression) effects of signal presence from both TPR and the task-evoked fMRI responses. These correlations were computed separately for each individual subject and then tested against 0 across the group (see section *Statistical comparisons*).

To compute the 'combined neuromodulatory brainstem signal' (*Figure 9*, *Figure 9—figure supplement 1*), we performed a multiple linear regression against TPR for the following ROIs: LC region, VTA, SN, BF-sept and BF-subl (see section *Definition of regions of interest*). This yielded a subject-specific set of weights per ROI that maximized the correlation to TPR. We then used these weights to compute, for each subject, a linear combination of the single-trial ROI-responses. The resulting signal was z-scored, with the purpose of using it as a predictor variable within the DDM framework

(see section *Analysis and modeling of choice behavior*). Likewise, the 'combined colliculi signal' was computed based on a linear combination of responses in SC and IC.

## Quantification of orientation-specific responses in early visual cortex

Orientation-specific response to the low-contrast target gratings in visual cortical areas V1-3 (*Figure 5B*; see section *Definition of regions of interest*) were computed using multi-voxel pattern analysis with a leave one out cross-validation procedure, as follows. First, per scanning session, we selected the 100 voxels with the strongest orientation preference from the unconstricted 'center' sub-region that corresponded retinotopically to the target stimulus. To this end, we selected the 50 most positive and 50 most negative t-values for the comparison between clockwise (CW) vs. counter-clockwise (CCW) stimuli on signal+noise trials:

$$t = \frac{\bar{x}_{cw}}{sem_{cw}} - \frac{\bar{x}_{ccw}}{sem_{ccw}},$$
(3)

where subscripts *cw* and *ccw* were clockwise and counter-clockwise target signals, respectively $\bar{x}_{cw}$ and $\bar{x}_{ccw}$ were the two sample means, and $sem_{cw}$ and $sem_{ccw}$ were the two corresponding standard errors. Second, the responses of the selected 100 voxels were arranged as vectors, each of which constituted a multi-voxel response pattern for one trial. Third, these single-trial response patterns were separately averaged across counter-clockwise and across clockwise trials, leaving out one trial per iteration. Fourth, a 'template pattern' was constructed as the difference between the mean counter-clockwise and clockwise response patterns. Fifth, the pattern of fMRI responses of the remaining trial (including noise trials) was correlated with the template pattern (on CW blocks), or with the inverted template (on CCW blocks). This procedure yielded a correlation coefficient per trial, which quantified the strength of the orientation-specific component of the population response of V1-V3.

## Quantification of choice-specific responses: univariate approach

We used two approaches to identify choice-encoding regions of interest (ROIs) and to quantify choice-specific signals therein. First, univariate logistic regression of binary choice against task-evoked fMRI responses (*Figure 7A*, and *Figure 7—figure supplement 1A,B*) described in this section; second, 'searchlight decoding' of choice based on multivariate local fMRI response patterns (*Figure 7B*, and *Figure 7—figure supplement 1C,D*), which is described in the subsequent section.

In the first approach, the following logistic regression model was fitted to the single-trial evoked fMRI responses of each voxel:

$$P(yes) = \frac{exp(\beta_0 + \beta_1 A)}{1 + exp(\beta_0 + \beta_1 A)}$$
(4)

where *P(yes)* was the predicted probability that the subject made a 'yes'-choice for a given value of A (in the model, a binary vector describing the choice identity (1,'yes'; 0, 'no') served as the dependent variable), *A* was a vector of fMRI response amplitudes, and $\beta_0$ and $\beta_1$ were the parameters of the fit. The parameters $\beta_0$ and $\beta_1$ were adjusted by an optimization routine to fit the measured probability of subjects' choices across all trials. The voxel was assigned the slope parameter, $\beta_1$, which quantified the link between the task-evoked fMRI responses of that voxel and the subject's behavioral choices. Performing this analysis for all voxels yielded a map of choice-specific activity for each subject. We performed the analysis separately for signal+noise trials and for noise trials and averaged the resulting maps. That way, the effect of choice was isolated, discarding potential differences between signal+noise and noise trials.

In the main task, 'yes'- and 'no'-choices were mapped onto motor responses with different hands. We therefore computed the logistic regression not only for each voxel's fMRI response, but also for the lateralization – that is, the difference between a voxel's response and its homotopic counterpart in the contralateral hemisphere. The mapping between the 'yes'- vs.'no'-choice and response hands was counterbalanced across subjects. Thus, the lateralization reflected the effector-specific activity (left vs. right hand button presses). Because in any given session 'yes'- and 'no'-choices were mapped onto button presses with different hands, the lateralization was a proxy of the choice-specific activity, encoded in the format of a plan to act (e.g., see [*Gold and Shadlen, 2007*]). Before

computing the lateralization, each voxel was flipped with its homotopic counterpart in the contralateral hemisphere for half of the subjects, for whom the mapping was: 'yes' -> right hand and 'no' -> left hand. The lateralization was then expressed with respect to the hand for 'yes' for all subjects (*Donner et al., 2009*). Consequently, lateralization values could be pooled across subjects without cancellation. The result of this analysis was visualized on the right hemisphere (*Figure 7A*, and *Figure 7—figure supplement 1A,B*).

We computed choice-specific fMRI response lateralization for a number of ROIs that were symmetric between the two hemispheres (see section *Definition of regions of interest*; *Figure 7A*, and *Figure 7—figure supplement 1A,B*). First, per scanning session, we selected the 50 voxels from the hemisphere contralateral to the hand used for reporting 'yes' with the most positive t-values for the comparison between 'yes'- vs. 'no'-choices (same as *Equation 3*, but now taking the difference of 'yes'- and 'no'-trials), and vice versa for the ipsilateral hemisphere. This adds up to a total of 100 voxels. Second, we separately averaged the single-trial responses across each of these two groups of voxels. Third, we computed the single-trial lateralization as the difference between the mean fMRI response of the 'yes' and the 'no' voxel group. Performing this analysis for each trial enabled computing single-trial correlations of fMRI lateralization values with TPR and behavioral choice, respectively.

To compute the 'combined lateralization signal' (*Figure 7D,E*), we fitted multiple logistic regression of choice on choice-specific responses in aIPS1 and IPS/PostCeS, and obtained the weighted sum of these responses that maximized the correlation to choice.

## Quantification of choice-specific responses: multivariate (searchlight decoding) approach

We used searchlight multi-voxel pattern classification to identify additional cortical regions which might encode the choice in other formats (e.g., more fine-grained patterns [*Hebart et al., 2016*, *2012*]) than the coarse hemispheric lateralization of response amplitudes. We used the LIBSVM implementation of a linear support vector machine (*Chang and Lin, 2011*), with a standard cost value of c = 1. A sphere of voxels (i.e. 'searchlight') was selected around a given voxel with a radius of 10 mm (435 voxels). From these voxels, the task-evoked fMRI responses were extracted, which made up the pattern vectors that were used for multivariate pattern classification of choices. We assigned each vector a label corresponding to the choice of the subject ('yes' vs. 'no'). The pattern vectors of all but one trial were then used to train a support vector machine to predict the category of the left-out pattern. After training, we validated the model by comparing the true label of the left-out pattern with the label predicted by the model. We repeated this train-test approach iteratively for each trial and calculated a mean cross-validated precision score across all trials for this searchlight. This precision score was assigned to the center voxel of the searchlight sphere. This procedure was repeated iteratively, for all voxels in the brain, yielding a map of precision scores for each subject. We performed the choice decoding analysis separately for signal+noise and noise trials and averaged the resulting maps. This combined individual map was spatially smoothed (FWHM: 4 mm) for group-level analyses.

Choice-specific responses for ROIs defined in the searchlight analysis (see section *Definition of regions of interest*; *Figure 7B*, and *Figure 7—figure supplement 1C,D*) were computed as follows. First, per scanning session, we selected from each ROI the 100 voxels with the strongest choice preference. To this end, we selected the 50 most positive and 50 most negative t-values for the comparison between 'yes'- vs. 'no'-choices (same as *Equation 3*, but now taking the difference of 'yes'- and 'no'-trials). Second, a 'template pattern' was constructed by averaging the task-evoked fMRI response of each selected voxel across all 'yes' trials minus the average on all 'no' trials. One trial was left out (leave one out cross-validation). Third, the pattern of fMRI responses of the remaining trial was correlated with the template pattern. This procedure yielded a correlation coefficient per trial, which quantified the sign and strength of the choice-specific pattern response. Again, performing this analysis for each trial enabled computing single-trial correlations of choice-specific pattern responses with TPR and behavioral choice, respectively.

To compute the 'combined searchlight signal' (*Figure 7D,E*), we fitted multiple logistic regression of choice on choice-specific responses in IPL, SPL, aIPS2, pIns, PreCeS/IFG and MFG, and obtained the weighted sum of these responses that maximized the correlation to choice. Although trial-to-trial

variations in RT had been removed (via linear regression) from the fMRI response of each individual voxel (see above), there was a small, but statistically significant correlation between the 'combined searchlight signal' and trial-by-trial RTs (group average r = 0.027, p=0.004, permutation test). However, this correlation was significantly smaller than the correlation between the 'combined searchlight signal' and behavioral choice (group average difference in correlation: Δr = 0.481, p<0.001). Thus, the multivariate choice decoder primarily decoded trial-to-trial variations in choice rather than in RT.

## Receiver-operating characteristic (ROC) analysis of stimulus- and choice-specific responses

We used ROC-analysis (*Green and Swets, 1966*) to quantify the reliability of the orientation- and choice-specific cortical responses at the single-trial level (*Figure 5—figure supplement 1*, *Figure 7*, and *Figure 7—figure supplement 1*). The ROC index ranged between 0 and 1, and quantified the probability with which one could predict the experimental variable of interest (in our case: signal presence or behavioral choice) based on responses measured during individual trials. An index of 0.5 implied chance level prediction. To remove effects of choice when predicting the stimulus, we averaged ROC indexes for signal presence obtained separately on 'yes' and 'no' trials (*Figure 5—figure supplement 1*). Similarly, to remove effects of stimulus when predicting choice, we averaged ROC indexes for choice obtained separately on signal+noise and noise trials (*Figure 7*, and *Figure 7—figure supplement 1*).

## Analysis of stimulus independent TPR effect on cortical signals

We used linear regression analysis to evaluate the hypothesis that the TPR-linked modulation of choice-specific responses is intrinsic, that is, if the modulation remained when factoring out effects of the physical evidence (*Figure 7*). We obtained the residual choice-specific (combined) responses by removing (via linear regression) variations due to stimulus (binary vector describing the stimulus identity (1, signal+noise; 0, noise).

Across 5 bins, defined by TPR, we then fitted the following linear model to the residual choice-specific responses:

$$\mathrm{Ctx} = \beta_0 + \beta_1 \mathrm{TPR} \tag{5}$$

where *Ctx* was a vector of mean residual (with effects of stimulus removed, see above) choice-specific responses per bin, *TPR* was a vector of mean task-evoked pupillary responses per bin, and $\beta_0$ and $\beta_1$ were the free parameters of the fit.

### Statistical comparisons

We used nonparametric paired permutation tests to test for significant differences between behavioral estimates (*Figure 2*, *Figure 2—figure supplement 1*, *Figures 3* and *4*, and *Figure 4—figure supplement 2*), task-evoked fMRI responses (*Figure 7*, *Figure 7—figure supplement 1*, *Figure 8*, and *Figure 8—figure supplement 1*), stimulus- / choice-predictive indices (*Figure 5—figure supplement 1*, *Figure 7*, and *Figure 7—figure supplement 1*), and regression beta weights / coefficients (*Figures 1*, *2*, *Figure 2—figure supplement 1*, *Figures 3*, *7*, *8*, *Figure 8—figure supplement 1*, *Figure 9*, and *Figure 9—figure supplement 1*) from different trial categories, or to test them against zero (against 0.5 in the case of stimulus- / choice-predictive indices). Statistical tests were performed at the group level, using the individual subjects' mean parameters as observations. For each comparison, we randomly permuted the labels of the observations (e.g., the regressor label of the beta estimates), and recalculated the difference between the two group means (10,000 permutations). The p-value was the fraction of permutations that exceeded the observed difference between the means.

We used nonparametric permutation tests within the FSL Randomise implementation to test cluster-corrected task-evoked fMRI responses against 0 (*Figure 5*), linear regression coefficients against 0 (*Figure 8*), logistic regression beta weights against 0.5 (*Figure 7*, *Figure 7—figure supplement 1*), and searchlight classification precision scores against 0.5 (*Figure 7*, *Figure 7—figure supplement 1*). Randomise implemented 10,000 randomly generated permutations of the data to perform a Monte Carlo-style permutation test. This procedure was robust with respect to inflated false-

positive rates (*Eklund et al., 2016*). In the majority of cases, we used a cluster correction threshold of p<0.01. For the logistic regression of binary choice against lateralized task-evoked fMRI responses (*Figure 7*, *Figure 7—figure supplement 1*), and in case of the brainstem TPR correlation (*Figure 8F*; accounting for the comparably low sensitivity of fMRI measurements in the brainstem), we used a cluster correction threshold of p<0.05.

## Definition of regions of interest (ROIs)

The analyses of fMRI signals focused on a number of ROIs, which were defined in each individual brain using a variety of criteria described in this section.

### Brainstem nuclei

We defined the following brainstem nuclei (*Figure 8*, and *Figure 8—figure supplement 1*) based on anatomical criteria.

### Locus coeruleus (LC)

The LC was delineated, separately for each subject and scanning session, by means of a specific (TSE) high-resolution structural MRI scan (*Figure 8A*, and *Figure 8—figure supplement 1A*). The LC, as some other brainstem structures, has increased neuromelanin concentration and can thus be identified in the TSE scans as two bilateral local maxima of elevated signal intensity within the mid-brain tegmentum and pons, at the floor of the fourth ventricle (*Keren et al., 2015*, *2009*; *Shibata et al., 2007*). We delineated LC voxels in the slice with the clearest signal intensity elevation in the designated region and in the one slice above and one below. The ROI definition was initially performed by the first author and then independently verified by another author. For the analysis of functional LC responses, we transformed the individually delineated high-resolution LC ROI to subject- and session-specific EPI space (trilinear affine transformation, using FSL FLIRT) after first (i) co-registering the high-resolution partial field-of-view TSE scan and whole-brain TSE scan (with 6 degrees of freedom), (ii) co-registering the whole-brain TSE scan and the T2-weighted anatomical scan (12 degrees of freedom), and (iii) concatenating the estimated transformation matrices. We then determined the minimum number of EPI voxels (obtained at lower resolution) for which the probability of the anatomical LC ROI contained in this EPI voxel was larger than 0 within each of the 14 subjects. The average number of voxels with a non-zero probability of containing the anatomical LC ROI was 18 voxels, and the smallest number across the group was 12. We therefore included for each subject the 12 voxels with the largest probability of containing the LC. The LC region time series was computed as the average of the time series from these 12 voxels, weighted by their probability values. For comparison, we used the same procedure, including the same number of voxels, for the other brainstem regions. For the most spatially specific definition of the LC region afforded by our measurements we computed a weighted average across the two voxels with the largest probability of containing the LC (*Figure 8D*).

### Superior and inferior colliculi (SC and IC)

We manually delineated the SC and IC based on the anatomical standard (MNI) brain as the two elevated 'hills' at the dorsal part of the midbrain.

### Other brainstem nuclei

We used probabilistic atlases in anatomical standard (MNI) space to delineate the following other brainstem nuclei: ventral tegmental area (VTA), substantia nigra (SN) (*Murty et al., 2014*), and two divisions of the basal forebrain (BF): one that includes cell groups within the septum and the horizontal limb of the diagonal band (BF-sept), and one that includes the sublenticular part of the basal forebrain (BF-subl). The probabilistic BF atlases (*Zaborszky et al., 2008*) were obtained from the SPM anatomy toolbox (*Eickhoff et al., 2005*). As we did for the LC, we transformed these anatomical ROIs to subject- and session-specific EPI space and thresholded the resulting masks such as to include only the 12 voxels with the largest probability of containing the particular structure. Likewise, ROI-level time series were computed as the average of the time series from these 12 voxels, with the weights corresponding to each voxel's probability value.

### Anterior cingulate cortex

We parcellated the cortex with the Freesurfer/Destrieux atlas (*Destrieux et al., 2010*), and obtained anterior cingulate cortex (ACC) masks by transforming the label G_and_S_cingul-Mid-Ant_d to sub-ject- and session-specific functional space (with Freesurfer mri_label2vol tool).

### Visual cortical areas V1-V3

Stimulus-responsive and surrounding sub-regions of early visual cortical areas V1-V3 (*Figure 5*) were defined in two steps. First, the boundaries between these visual cortical areas were identified by reti-notopic mapping via population receptive field imaging (i.e., quantifying, for each voxel, the pre-ferred polar angle and eccentricity) (*Dumoulin and Wandell, 2008*). To this end, in a separate scanning session, we presented moving bar apertures consisting of a large number of randomly ori-ented Gabor patches (100% contrast). The width of the bar subtended 25% of the stimulus radius. Four bar orientations (0°, 45°, 90°, 135°) and two different motion directions for each bar were used, yielding a total of eight different bar configurations within a given scan. Subjects fixated at the cen-ter of the screen, and pressed a button with their right index finger when the fixation dot changed color (from red to green) at random time intervals. Second, using a localizer run in each session of the main experiment, we identified the sub-regions of V1-V3 that corresponded retinotopically to the stimulus judged in the main experimental (yes-no) task. To this end, full-contrast gratings were presented at the same position as the stimuli in the main experimental task, in periodic alternation with central fixation of an otherwise blank screen (block duration: 12 s). *Figure 5A* shows the map of localizer responses in one example subject.

### Choice-specific cortical regions

Choice-specific ROIs (*Figure 7*, and *Figure 7—figure supplement 1*) were defined as follows: uni-variate logistic regression on lateralized task-evoked fMRI responses and multivariate searchlight decoding (see section *Analysis of MRI data*). Group-level cluster correction yielded the conjunction maps shown *Figure 7A,B*, and *Figure 7—figure supplement 1A–D*. Performing these analyses sep-arately for each of the two scanning sessions allowed us to identify robust choice-specific ROIs as clusters of voxels, which exhibited statistically significant choice-specific activity within *both* indepen-dent sessions (conjunction maps in *Figure 7A,B*).

The first approach yielded the following choice-specific ROIs: hand region of primary motor cor-tex in the central sulcus (M1), posterior parietal cortex, in the junction of the intraparietal sulcus and the postcentral sulcus (IPS/PostCeS), and the anterior intraparietal sulcus (aIPS1) (*Figure 7A*, and *Figure 7—figure supplement 1A,B*).

The second approach yielded the following choice-specific ROIs: the superior parietal lobule (SPL), the inferior parietal lobule (IPL), a second region within the anterior intraparietal sulcus (aIPS2), the posterior insula (pIns), the junction of the precentral sulcus and the inferior frontal gyrus (Pre-CeS/IFG), and the medial frontal gyrus (MFG) (*Figure 7B*, and *Figure 7—figure supplement 1C,D*).

## Data and code sharing

The data are publicly available on Figshare (*de Gee et al., 2017a*, https://doi.org/10.6084/m9.fig-share.4806562; *de Gee et al., 2017b*, https://doi.org/10.6084/m9.figshare.4806559). Analysis scripts are publicly available on Github (*de Gee, 2017a*, https://github.com/jwdegee/2017_eLife (with a copy archived at https://github.com/elifesciences-publications/2017_eLife); *de Gee, 2017b*, https://github.com/jwdegee/2014_PNAS).

## Acknowledgements

We thank Daniel Lindh for help with the data collection of the main experiment. We thank Matthew McGinley for providing the stimuli, and Daniëlle Rijkmans, Guusje Boomgaard and Christopher David Riddell for help with the data collection for the auditory detection task. We thank Konstantinos Tset-sos, Angela Yu, David J Heeger, Birte U Forstmann and Todd Hare for discussion. This research was supported by the German Research Foundation (DFG, SFB 936/Z1 and DO 1240/3–1, to THD), the European Union Seventh Framework Programme (FP7/2007-2013) under grant agreement no.

604102 (Human Brain Project, to THD), and the European Research Council (ERC) under grant agreement no. 283314-NOREPI (to SN).

## Additional information

### Funding

| Funder | Grant reference number | Author |
|---|---|---|
| European Research Council | 283314-NOREPI | Sander Nieuwenhuis |
| Deutsche Forschungsgemeinschaft | SFB 936/Z1 | Tobias H Donner |
| Deutsche Forschungsgemeinschaft | DO1240/3-1 | Tobias H Donner |
| Seventh Framework Programme | 604102 | Tobias H Donner |

The funders had no role in study design, data collection and interpretation, or the decision to submit the work for publication.

### Author contributions

JWdG, Conceptualization, Formal analysis, Investigation, Writing—original draft, Writing—review and editing; OC, NAK, Investigation, Writing—review and editing; TK, SN, Resources, Writing—review and editing; THD, Conceptualization, Resources, Supervision, Writing—original draft, Writing—review and editing

### Author ORCIDs

Jan Willem de Gee, http://orcid.org/0000-0002-5875-8282
Olympia Colizoli, http://orcid.org/0000-0001-5288-2437
Niels A Kloosterman, http://orcid.org/0000-0002-1134-7996
Tomas Knapen, http://orcid.org/0000-0001-5863-8689
Sander Nieuwenhuis, http://orcid.org/0000-0003-2418-3879
Tobias H Donner, http://orcid.org/0000-0002-7559-6019

### Ethics

Human subjects: All subjects gave written informed consent, and consent to publish. The ethics committee of the Psychology Department of the University of Amsterdam approved the experiments (Id's: 2014-BC-3406; 2015-BC-4613; 2016-BC-6842).

## Additional files

### Major datasets

The following dataset was generated:

| Author(s) | Year | Dataset title | Dataset URL | Database, license, and accessibility information |
|---|---|---|---|---|
| Jan Willem de Gee, Olympia Colizoli, Niels A Kloosterman, Tomas Knapen, Sander Nieuwenhuis, Tobias H Donner | 2017 | Data set: Dynamic modulation of decision biases by brainstem arousal systems | https://doi.org/10.6084/m9.figshare.4806562 | Available at figshare under a CC0 Public Domain licence (https://figshare.com/) |

The following previously published dataset was used:

| | | | | Database, license, and accessibility |
|---|---|---|---|---|

| Author(s) | Year | Dataset title | Dataset URL | information |
|---|---|---|---|---|
| de Gee JW, Knapen T, Donner TH | 2017 | Data set: Decision-related pupil dilation reflects upcoming choice and individual bias | https://doi.org/10.6084/m9.figshare.4806559 | Available at figshare under a CC0 Public Domain licence (https://figshare.com/) |

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
