## [Decision Letter]

Thank you for submitting your article "Dynamic Modulation of Cortical and Behavioral Decision Biases by Brainstem Arousal Systems" for consideration by *eLife*. Your article has been favorably evaluated by Sabine Kastner (Senior Editor) and three reviewers, one of whom, Klaas Enno Stephan (Reviewer #1), is a member of our Board of Reviewing Editors. The following individual involved in review of your submission has agreed to reveal their identity: Micah Allen (Reviewer #3).

The reviewers have discussed the reviews with one another and the Reviewing Editor has drafted this decision to help you prepare a revised submission.

Summary of manuscript:

This study examines biases in human perceptual decision-making processes, testing the hypothesis that they may arise from fluctuations in the activity of neuromodulatory brainstem nuclei, such as the locus coeruleus, which may correspond to variations in arousal. Participants performed a simple forced choice perceptual detection task in the MRI scanner while their pupil responses were tracked. The authors report that larger peri-response pupil sizes were associated with reduced bias, and that pupil size variation was reflected by activity in prefrontal and parietal cortex, the basal forebrain, and noradrenergic and dopaminergic brainstem nuclei. They conclude that phasic arousal signals explain a significant portion of choice variability. While the authors have reported the behavioral effects previously (de Gee et al., 2014, PNAS), the current paper goes an important step further in examining potential neurobiological mechanisms.

Summary of reviews:

Overall, all reviewers agreed that the paper represents an important step forward in understanding possible neurobiological determinants of choice variability. However, they also thought that the paper was densely written and not easy to read, and they identified a number of issues that would need to be addressed in a revision of the paper.

The policy of the journal is to provide you with a single set of comments which reflect the consensus view amongst reviewers. These comments can be found below and must be addressed convincingly. We hope that you will find these comments helpful to further improve the paper.

1) The paper is densely written and not easy to read. Given that the *eLife* does not impose length restrictions, you would have the liberty to explain your rationale, analyses and findings in a less condensed way. We think it would be very helpful for most readers and would enhance the reception of your paper if you could consider this and reformulate the text where you think it is appropriate.

2) The fMRI analysis of effects of phasic arousal (as indexed by TPR) on sensory responses in visual cortex (subsection “Phasic arousal predicts a reduction of accumulation bias”) appears to be unnecessarily complicated. Would it not be much simpler (and arguably more direct) to test for TPR x stimulus interactions, using a standard GLM?

3) A similar concern applies to the next section, the fMRI analysis of phasic arousal on decision-making (subsection “Phasic arousal predicts selective changes in frontal and parietal decision signals”). The analysis presented does not refer to computational variables (from the drift diffusion model) but tests for an effect of TPR on decision-making in an indirect way (differences in lateralisation of activity between low and high TPR trials). Against this background, the conclusion that "[…] the selective effect of phasic arousal on behavioural bias is mediated by a selective modulation of cortical *decision signals*" (our emphasis) seems a little overstated; testing for interactions between TPR x drift rate interactions would have been more convincing. Furthermore, it would help if you clarified what exactly you mean by "selective" (this word occurs frequently throughout the paper) and how in this analysis you operationalise "lateralisation".

4) The analysis of association between TPR and activity in brainstem nuclei is compelling as it does not restrict itself to the locus coerulus. Having said this, it is not clear how/whether you corrected for multiple comparisons – could you please clarify?

5) Several aspects of the mediation (structural equation modeling) analysis are not clear. First, it is not clear why the structural model presented is the most obvious choice, and it would be helpful to see a model comparison (e.g., based on BIC) against at least two alternative models: (i) a model that does not allow for the indirect path (i.e. a model lacking the path from TPR to *Ctx*); (ii) a model without stimulus input into Choice. Second, what exactly is the "cortical response" in *Ctx*? Third, where does the "predicted probability that the subject made a yes-choice" (subsection “Mediation analysis of TPR effect on cortical signals and behavior”) come from?

6) Psychologically it is unclear whether there might be alternative interpretations of what the variations in pupil size reflect. Given that activity fluctuations in multiple neuromodulatory nuclei show a relation to TPR, is a strong conclusion/interpretation in terms of a "phasic arousal signal" warranted? This may be an obvious interpretation for noradrenaline, but is this an appropriate interpretation for dopamine and acetylcholine? For example, could something like motor confidence, (violations in the) expectation of being correct, or simply fluctuations in attention represent alternative interpretations?

7) It would be helpful if the relation of pupil responses to response times could be illustrated. Relatedly, could you verify that you did not decode differences in response times in the multivariate choice analysis?

8) Although it used to be standard practice for the authors of psychophysics papers to be subjects too, one wonders whether in this case, the inclusion of authors as subjects may have biased the findings, given that they knew the results of the previous paper. Do the results quantitatively remain the same when excluding the two authors? Relatedly, one subject performed significantly more trials than the other subjects (640 instead of 400-480 trials). Please specify whether this was one of the authors and demonstrate that this subject did not bias the findings.

9) A sense factor of 3 is relatively high, and one might be concerned that residual aliasing or noise enhancement affected the data. Could you please address this?

10) Model comparison (subsection “Phasic arousal predicts a reduction of accumulation bias”): do the DIC values reported reflect an overall group result, i.e., was the model treated as a fixed effect in the population? If so, did you verify that this result was not driven by a single or few outliers (a random effects analysis would protect against this)? Do AIC or BIC provide for converging conclusions?

---

## [Author Response]

*[…] 1) The paper is densely written and not easy to read. Given that the eLife does not impose length restrictions, you would have the liberty to explain your rationale, analyses and findings in a less condensed way. We think it would be very helpful for most readers and would enhance the reception of your paper if you could consider this and reformulate the text where you think it is appropriate.*

We are happy with the opportunity to expand the text and present the rationale, analyses and findings in a less condensed way. We agree that this substantially aids the comprehensibility of the paper. We have now revised the entire text accordingly. In so doing, we have focused in particular on the following two points, which we felt did not sufficiently come across in the original submission:

A) The difference between the drift diffusion model parameters “starting point”, “drift rate”, and “drift criterion” (see Results section “Phasic arousal predicts a reduction of accumulation bias*”*, updated Figure 4, and the new Figure 4—figure supplement 1). In discussions with several colleagues, we have realized that the drift criterion parameter does not seem to be widely known, even among researchers using the drift diffusion model in their own research. The reason seems to be that the model is most commonly fitted to reaction time (RT) distributions for correct and error trials, after collapsing across choices/responses (so-called “accuracy coding”). Only the relatively few studies in the literature that used the model to unravel mechanisms underlying choice biases have estimated drift criterion. This requires fitting RT distributions split by choice/response, while coding the stimulus category of each trial (so-called “stimulus coding”). We now explain this parameter in more detail and present simulations that show how the model can tease apart biases due to shifts in starting point from biases due to changes in drift criterion (Figure 4—figure supplement 1).

B) Rationale for our focus on the stimulus- and choice-specific fMRI responses. We think that these specific signal components are more useful for linking the fMRI data to the algorithmic correlates of phasic arousal than the overall levels of fMRI responses. The reasoning is as follows. The decision variable in both signal detection theory and drift diffusion model is a signed quantity that encodes the relative support in favor of one over the other choice. Thus, neural correlates of these variables should also be signed, and specific for one or the other choice. This is the same rationale that underlies the vast majority of the highly influential single-unit physiology studies of perceptual choice (work by leading labs). We do realize that this rationale has, at least so far, not been widely used in human neuroimaging studies of perceptual choice (although there are some exceptions). Thus, we now elaborate on this point in more detail at several points: Results sections “Phasic arousal does not boost sensory responses in visual cortex*”* and *“*Phasic arousal modulates choice-specific signals in frontal and parietal cortex*”*. See also our reply to issues #2 and #3 below.

*2) The fMRI analysis of effects of phasic arousal (as indexed by TPR) on sensory responses in visual cortex (subsection “Phasic arousal predicts a reduction of accumulation bias”) appears to be unnecessarily complicated. Would it not be much simpler (and arguably more direct) to test for TPR x stimulus interactions, using a standard GLM?*

Thank you for this helpful suggestion, and for pointing us to the complexity of the description. To address this issue, we have completely changed the figure and the corresponding text in the Results section “Phasic arousal does not boost sensory responses in visual cortex”: We now show these interactions between stimulus and TPR in the main Figure 5 (panel B, focusing on stimulus-specific response component in visual cortex) as well as in a new main Figure 6 (panel D, map of the above interaction in the overall task-evoked fMRI signal across cortex). Neither analysis revealed a significant interaction between stimulus and TPR.

We assume that part of the complexity that this comment refers to stemmed from our shift in focus from the overall fMRI response in retinotopic visual cortex to the stimulus-specific component of the multivariate fMRI response. We think the latter, but not the former, is crucial for addressing the question whether phasic arousal modulates the *sensory encoding* of the target. Figure 5 shows that our stimulus-specific signal taps into that encoding, in the sense that it reliably discriminates between presence and absence of the target signal. The original version of the figure showed, in addition, that the target presence is *not* encoded in the overall fMRI response. To simplify this part of the Results, we have now dropped the overall fMRI responses from Figure 5. Instead we now show maps of the cortical distribution of the TPR- and stimulus-effects, and their (absent) interaction, in a new main Figure 6.

The absence of stimulus encoding in the overall fMRI response in visual cortex may seem surprising, although it has antecedents in the literature on near-threshold detection tasks (Ress et al., 2000; Ress and Heeger, 2003) and relates to a bigger issue pertaining to the neurophysiological underpinnings of the BOLD signal (Cardoso et al., 2012; Logothetis, 2008). For the sake of simplicity and focus on the main theoretical questions, we refrained from elaborating on these methodological issues in the present paper (although we do find them important and interesting). Instead, we simply moved on to exploit clearly established insights from single-unit physiology to extract a component of the fMRI signal that actually does reliably encode the target stimulus: the orientation-specific component of the population response, as shown in Figure 5. As alluded to in our reply to issue #1, we have now clarified our rationale and explicitly defined “stimulus-specific”:

“Because the majority of visual cortical neurons encoding stimulus contrast are also tuned to stimulus orientation, orientation-tuning could serve as a “filter” to separate the cortical stimulus response from stimulus-unrelated signals. […] Consequently, we henceforth refer to this component as the “stimulus-specific response”.

We then went on to test if TPR and stimulus presence exhibit interacting effects on this stimulus-specific response (Figure 5). We hope and expect that this re-written sub-section of the Results is easier to follow than the original presentation.

*3) A similar concern applies to the next section, the fMRI analysis of phasic arousal on decision-making (subsection “Phasic arousal predicts selective changes in frontal and parietal decision signals”). The analysis presented does not refer to computational variables (from the drift diffusion model) but tests for an effect of TPR on decision-making in an indirect way (differences in lateralisation of activity between low and high TPR trials).*

We have also clarified our rationale behind the approach to cortical choice signals. The insight that the increase of “yes”-choices (and the resulting bias reduction) was due to a bias reduction in the evidence accumulation process itself (Figure 4) led to the strong prediction: “[…] the observed effect of TPR on choice bias (criterion, drift criterion) predicted a directed shift (towards “yes”) in neural signals encoding subjects’ choices, in downstream cortical regions”. We went on to test that prediction, establishing that phasic pupil-linked arousal predicts changes in choice-specific frontal and parietal decision signals (Figure 6). Indeed, we used separate analyses to assess if and how TPR modulates (i) computational variables of the drift diffusion model and (ii) choice-encoding cortical signals.

We think that identifying and carefully characterizing choice-specific cortical signals was crucial before correlating these signals to TPR. Further, we think that the single-trial correlation with specific (i.e., “yes” vs. “no”) choices was exactly the right approach for doing so. Even if this did not involve explicit correlations to DDM parameters, our analysis approach was constructed to identify signals whose functional properties have a straightforward correspondence to the decision variable postulated by models like the DDM, in the sense that their signed values correspond to a tendency for either of the two choices at stake. Please note that our current approach is conceptually analogous to approaches for identifying decision-related activity in single neurons. Our approach is based on our previous neuroimaging work, which has translated the single-unit approaches to human neuroscience (de Lange et al., 2013; Donner et al., 2009; Hebart et al., 2012; 2016). The main question of this part of our current paper was, if and how choice-specific cortical signals are modulated by TPR. We feel this question is conclusively answered by the analyses presented.

That said, the comment did motivate us to perform a new, model-based analysis of another part of our fMRI data, namely the brainstem nuclei. Here, we model the influence of the neuromodulatory brainstem signals on the drift at the single-trial level, to directly test the hypothesis that phasic neuromodulatory signals (but not the responses of the superior/inferior colliculi) suppress biases inherent in the single-trial drift of the decision variable. This analysis is presented in the new Figure 9, and we believe that it substantially strengthens and specifies the core conclusion of the paper.

*Against this background, the conclusion that "[…] the selective effect of phasic arousal on behavioural bias is mediated by a selective modulation of cortical decision signals" (our emphasis) seems a little overstated; testing for interactions between TPR x drift rate interactions would have been more convincing.*

Given the issues discussed in detail under issue #5 below, we agree that the mediation statement was overstated. We have now toned down this statement throughout. For example, at the end of the Introduction we now write:

“Using fMRI for one of these tasks revealed that the bias reduction was accompanied by a modulation of choice-encoding response patterns in prefrontal and parietal cortex.”

Furthermore, it would help if you clarified what exactly you mean by "selective" (this word occurs frequently throughout the paper).

We used “selective” in analogy with the terminology from single-unit physiology, in which neuronal responses are being classified as “selective” for a particular stimulus, cognitive, or movement variable, based on analyses of neuronal tuning curves, mutual information, or ANOVAs. However, we realized that we used the attributes “selective” and “specific” interchangeably throughout our manuscript, which may have been confusing. To eliminate this confusion, we decided to only use “specific”, and to define this upon first use (see also our replies to issues #2 and #3). We have now defined “stimulus-specific” as follows:

“The orientation-specific response component also reliably discriminated between signal+noise and noise trials on a single-trial basis (Figure 5—figure supplement 1). Consequently, we henceforth refer to this component as the “stimulus-specific response”.

Likewise, we have defined “choice-specific” as follows:

“Here, we use the term “choice-specific” to refer to fMRI-signals that reliably discriminated between subjects’ choice (“yes” vs. “no”).”

*And how in this analysis you operationalise "lateralisation".*

We now describe in more detail how lateralization was operationalized in the Materials and methods section “Quantification of choice-specific responses univariate approach*”*:

“In the main task, “yes”- and “no”-choices were mapped onto motor responses with different hands. […] The result of this analysis was visualized on the right hemisphere (Figure 7, and Figure 7—figure supplement 1).”

*4) The analysis of association between TPR and activity in brainstem nuclei is compelling as it does not restrict itself to the locus coerulus. Having said this, it is not clear how/whether you corrected for multiple comparisons – could you please clarify?*

The map of single-trial correlations between TPR and evoked fMRI responses (Figure 8) was indeed plotted after cluster-based multiple comparison correction (nonparametric permutation test within the FSL Randomise implementation; see also Materials and methods section “Statistical comparisons*”*).

The matrix of correlations between evoked brainstem fMRI responses (Figure 8) was, likewise, corrected for multiple comparison, now using false discovery rate (FDR). We have added this to the figure legend.

Significant clusters in the map of Figure 8 overlapped with five out of seven brainstem ROIs presented in Figure 8 panels H and I, and Figure 8—figure supplement 1 panels B and C (SC, LC, VTA, SN and BF-subl). Because the analyses in the above panels also quantified the same effect (the coupling between TPR and evoked fMRI responses) the statistics presented in these panels were not further corrected.

*5) Several aspects of the mediation (structural equation modeling) analysis are not clear.*

Thank you for raising this important issue. Running the analyses to address your point made us realize a shortcoming of the analysis that we had not noticed before. After careful consideration, we have now decided to replace it with a simpler approach, which suffices to address the question we set out to test here. In what follows, we first elaborate on this decision, and then address your specific questions pertaining to the original analysis.

In the mediation analysis we had quantified the indirect path (*a • b*; see Figure 10), and tested this against 0. However, the choice-specific cortical signals (*Ctx*) were selected on the basis of strong correlation with choice. This implies that the coefficient *b* was constrained to consistently positive (and large) values in all subjects. Consequently, statistical significance of the term (*a • b*) was largely dependent on the inter-subject variability in *a*. This means that a simpler model assessing the same effect is a test of our former *a*-path. We now present this simpler analysis in the new Figure 7. It supports the conclusion that phasic arousal accounts for a significant component of the variability of choice-encoding response patterns in prefrontal and parietal cortex, over and above the evidence.

The analysis (as the analysis that we originally presented) does *not*, however, provide sufficiently strong support for the conclusion that the TPR or brainstem effects on decision bias are *mediated* by this modulation of cortical signals, as described below. Consequently, we have toned down this part of the conclusion. We even removed “cortical” from the title in order to focus the paper on what we think to be the main and novel insights: the computational and brainstem correlates of pupil-linked, phasic arousal. Overall, we think these changes have led to a fairer representation of the data, without diluting our key conclusions.

Author response image 1.(**A**) Left: Schematic of original mediation analysis. Ctx, choice-specific cortical response; stim, stimulus (signal+noise vs. noise). All arrows are regressions.Right: BIC values for different choice-specific cortical responses (independently used as the Ctx node). (**B**) same as panel A, but for alternative model #1 (without stimulus input into Choice). (**C**) same as panel A, but for alternative model #2 (that does not allow for the indirect path – that is, a model lacking the path from TPR to Ctx).**DOI:**
http://dx.doi.org/10.7554/eLife.23232.028

*First, it is not clear why the structural model presented is the most obvious choice, and it would be helpful to see a model comparison (e.g., based on BIC) against at least two alternative models: (i) a model that does not allow for the indirect path (i.e. a model lacking the path from TPR to Ctx); (ii) a model without stimulus input into Choice.*

We computed Bayesian information criterion (BIC) values for three different models (see also Figure 10):

The original model, as presented in former Figure 7;

An alternative model (AM#1) without stimulus input into Choice;

An alternative model (AM#2) that does not allow for the indirect path (that is, a model lacking the path from TPR to *Ctx*).

First, we compared our original model to AM#1. We found a difference in BIC for the “combined choice signal” of -92.57 (+/- 13.06 s.e.m.), and virtually the same numbers for the other choice-specific cortical responses. This corresponds to very strong evidence for our original model.

Second, we compared our original model to AM#2. We found a difference in BIC for the “Combined” response of 3.65 (+/- 0.73 s.e.m.), and virtually the same numbers for the other choice-specific cortical responses. This means we cannot decisively arbitrate between our original model and AM#2. Having said that, we would like to point out, that basic neurobiology favors our original model: pupil responses *cannot* affect choice directly, so there *must* be mediation via *some* signal in the brain. Nonetheless, the ambiguity inherent in the data with respect to this point favored dropping this aspect of the conclusion.

The corresponding models and their statistical tests and BIC values are shown below.

*Second, what exactly is the "cortical response" in Ctx?*

With *Ctx* we indicate any of the choice-specific cortical responses identified in Figure 7 (former Figure 6). These were separately entered in the mediation analysis, or linearly combined, in order to construct the “combined choice-specific signal”.

*Third, where does the "predicted probability that the subject made a yes-choice" (subsection “Mediation analysis of TPR effect on cortical signals and behavior”) come from?*

For this part of the regression model, we used logistic regression, modeling the probability of a binary response (“yes” vs. “no”) given particular values of the continuous predictors *TPR* and *Ctx*. We clarified this point in the Materials and methods section “Quantification of choice-specific responses: univariate approach”, in which we introduce the logistic regression approach for the first time.

*6) Psychologically it is unclear whether there might be alternative interpretations of what the variations in pupil size reflect. Given that activity fluctuations in multiple neuromodulatory nuclei show a relation to TPR, is a strong conclusion/interpretation in terms of a "phasic arousal signal" warranted? This may be an obvious interpretation for noradrenaline, but is this an appropriate interpretation for dopamine and acetylcholine? For example, could something like motor confidence, (violations in the) expectation of being correct, or simply fluctuations in attention represent alternative interpretations?*

Our focus on pupil-linked arousal is based on recent work establishing a close link between pupil diameter and “cortical state” (as defined based on a range of different physiological parameters) – these associations are strong and range from single-neuron membrane potentials to LFPs and brain-wide patters of fMRI responses (Eldar et al., 2013; McGinley et al., 2015a; Pisauro et al., 2016; Reimer et al., 2014; Vinck et al., 2015; Yellin et al., 2015). For example, McGinley et al. (2015) report coherence between pupil diameter and hippocampal ripple rate of about 0.8 (their Figure 1), and pupil diameter and neocortical membrane potential fluctuations of about 0.7 (their Figure 4). Vinck et al. (2015) report correlations of pupil diameter with LFP activity in δ (1-4 Hz) and γ (55-65 Hz) frequency bands of close to 1 (their Figure 2 and Figure 6). We have added a statement to the opening paragraph of the Results section that qualifies our use of the term “pupil-linked arousal”:

“We systematically quantified the interaction between pupil-linked arousal responses and decision computations, at the algorithmic and neural levels of analysis. We here operationalize “phasic arousal” as task-evoked pupil responses (TPR). This operational definition is based on recent animal work, which established remarkably strong correlations between non-luminance mediated variations in pupil diameter and global cortical arousal state (McGinley et al., 2015b).”

This clarifies that we use the above (physiologically defined) concept of arousal as *starting point* of our analysis, not as an *interpretation* of our findings. Given the well-defined neurophysiological correlates of (non-luminance mediated) changes in pupil diameter, we ask how those neurophysiological effects shape decision computations. In so doing, we remain deliberately agnostic about the psychological interpretation of pupil-linked arousal. While we are open to the idea that pupil dilation might correlate with psychological variables, such as those listed in your comment, we did not measure or operationalize those variables explicitly, and we do not see the evidence that would support a one-to-one mapping between particular neuromodulators and such variables. Therefore, we prefer stick to our operationally-defined and general term “pupil-linked arousal” in this paper.

*7) It would be helpful if the relation of pupil responses to response times could be illustrated.*

Thank you for the suggestion. We added an illustration of the relationship between TPR and response times (Figure 1—figure supplement 1), and refer to this panel in the Materials and methods section “Quantification of task-evoked pupillary responses (TPR)”.

*Relatedly, could you verify that you did not decode differences in response times in the multivariate choice analysis?*

Indeed, in the yes-no detection task, response time (RT) was smaller for “yes”- than for “no”-choices in the fMRI data set (median of median RTs: “yes”: 2035 ms, “no”: 2175 ms). Thus, one might wonder if RT might have contributed to our decoding analyses. This scenario seemed unlikely *a priori*, because we removed the effect of trial-to-trial variations in RT from each voxel’s fMRI responses. See the Materials and methods section “Quantification of task-evoked fMRI responses and correlation with TPR”:

“Trial-to-trial variations of each voxel’s task-evoked fMRI response due to RT were removed (via linear regression), to isolate fMRI response variations that were due to variations in the amplitude of the underlying neural responses (see section Analysis of task-evoked pupil responses).”

In addition, we have now verified that trial-to-trial variability in RT was only a minor (if any) factor contributing to our choice decoding results on the residual fMRI responses. In our version of multivariate pattern analysis, positive correlation coefficients corresponded (by construction) to stronger tendency towards a “yes”-choice (see Materials and methods section “Quantification of choice-specific responses: multivariate (searchlight decoding) approach”). At the same time, RTs were, on average, shorter for “yes”- than for “no”-choices. Thus, had RT contributed to the decoding of “yes” choices, the correlation between choice-specific responses and RT should have been *negative*. In contrast to this prediction, we observed a small, but statistically significant *positive* correlation between the combined “searchlight signal” and RT (group average r = 0.027, p = 0.004, permutation test). This indicates that, if anything, RT variations counteracted our choice decoding results.

Most importantly, the above correlation was negligible. Indeed, the correlation was substantially and significantly smaller than the correlation between the combined searchlight signal and behavioral choice (group average difference in correlation: Δr = 0.481, p < 0.001). The same qualitative pattern of results for choice-specific responses was evident for all the individual ROIs obtained through searchlight classification. In sum, the multivariate choice classifier primarily decoded choice rather than RT. This issue is now addressed in Materials and methods:

“Although trial-to-trial variations in RT had been removed (via linear regression) from the fMRI response of each individual voxel (see above), there was a small, but statistically significant correlation between the “combined searchlight signal” and trial-by-trial RTs (group average r = 0.027, p = 0.004, permutation test). […] Thus, the multivariate choice decoder primarily decoded trial-to-trial variations in choice rather than in RT.”

*8) Although it used to be standard practice for the authors of psychophysics papers to be subjects too, one wonders whether in this case, the inclusion of authors as subjects may have biased the findings, given that they knew the results of the previous paper. Do the results quantitatively remain the same when excluding the two authors? Relatedly, one subject performed significantly more trials than the other subjects (640 instead of 400-480 trials). Please specify whether this was one of the authors and demonstrate that this subject did not bias the findings.*

We share your general concern about including authors as subjects, and we generally refrain from doing so in most of the ongoing work in our lab. In the case of this particular study, we reasoned that the risk of biasing our results through the inclusion of authors as subjects would, however, be small. This is because we envisage the key process under study – phasic pupil-linked arousal – as rather automatic. Specifically, subjects’ control over their pupil responses during decisions, the trial-to-trial fluctuations of which we exploited in our study, seems negligible.

To completely rule out possible biases, we have now recomputed all 2^nd^ level statistics for the group, excluding three authors and the one subject who performed 640 trials (who was not an author). The results of this control analysis were qualitatively identical, except for two minor differences which are not essential for the central conclusions of this paper: (i) the correlation between TPR and the combined “lateralization signal” (after regressing out stimulus) was only marginally significant (p = 0.064; Figure 7); and (ii) the partial correlation between TPR and the BF-subl just fell short of statistical significance (p = 0.056; Figure 8). In both cases, the direction of the effect remained the same as for the complete set of 14 subjects, indicating that there is nothing special about these four subjects. The full set of main figures for N=10 is appended to the bottom of this response to reviewers, and we have added to the Materials and methods section *“*Subjects”:

"Three subjects were authors, and the remaining 12 subjects were naive to the purpose of the study. The results were unchanged when excluding the three authors (see *Author response*, online) and the one subject who performed three sessions (and more trials; see section *Behavioral tasks*) of the main experiment.”

We would also like to point out that there was an error in the original submission, which we have now corrected. Three, not two, of ours subjects were authors. The third “author-subject” (OC) joined the author list after we had composed the first draft, by contributing the data from the motion-discrimination control experiment (Figure 3), which we realized was important for generalizing our conclusions. We apologize for the confusion.

*9) A sense factor of 3 is relatively high, and one might be concerned that residual aliasing or noise enhancement affected the data. Could you please address this?*

We are aware that a sense factor of 3 is relatively high. This choice was governed by the need to image a large part of brain (brainstem and most of the cortex) while keeping spatial and temporal resolution (inverse of TR) comparably high. A relatively high image resolution was important for the brainstem analyses in particular, because its nuclei of interest are small (primarily the locus coeruleus). A relatively high temporal resolution (i.e., short TR) was important for the reliable quantification of single-trial fMRI response time courses, which formed the basis of all our statistical analyses.

That said, there is no indication that residual aliasing or noise enhancement were a matter of concern. First, we found no residual aliasing artifacts (i.e., folding) in our EPI data. Figure 11 shows example images from six of our subjects. These are representative for the images we analyzed in this data set.

Author response image 2.Raw EPI-images of six example subjects.The images depict the full field of view covered by our measurements. Aliasing artifacts occur when the dimensions of an object exceed the imaging field-of-view, but are within the area covered by the receiver coil. This “wrap-around artifact” is evident as a folding over of surrounding parts into the area of interest, and it is most severe along the phase-encode dimension. In our measurement, the phase-encode dimension was the anterior-posterior axis. No wrap-around artifacts are evident in any of the examples, nor in the other image data that were part of this data set.**DOI:**
http://dx.doi.org/10.7554/eLife.23232.029

Second, we verified that all our individual fMRI data were of sufficiently high signal-to-noise ratio to perform meaningful second-level analyses. To verify this, we have systematically assessed not only statistical parameter estimates, but also the actual fMRI response time courses for all brain regions of interest, at both, the group and single-subject level This verified well-behaved fMRI response time courses for all brainstem regions (Figure 8), as well as for all cortical regions under study. Response time courses of cortical regions were not shown in the figures, to keep the already long paper at a manageable size. We show the group average task-evoked response time courses for the cortical regions of interest in Figure 12. Again, these are well-behaved for all cortical regions, and the s.e.m. (shading) around the group average was small, indicating small across-subjects variability.

Author response image 3.Task-evoked responses across all trials.Regions of interest were pooled across hemispheres. Green bar, time course significantly different from zero (p < 0.05; cluster-corrected). Grey box, time window for computing scalar response amplitudes.**DOI:**
http://dx.doi.org/10.7554/eLife.23232.030

*10) Model comparison (subsection “Phasic arousal predicts a reduction of accumulation bias”): do the DIC values reported reflect an overall group result, i.e., was the model treated as a fixed effect in the population? If so, did you verify that this result was not driven by a single or few outliers (a random effects analysis would protect against this)?*

The model was not treated as a fixed effect in the population, but instead fitted with a hierarchical Bayesian approach. In the Materials and methods section “Analysis and modeling of choice behavior” we write:

“We fitted the model using the hierarchical Bayesian implementation of the HDDM toolbox (Wiecki et al., 2013) (version 0.6). The group distribution constrains individual subject parameter estimates, with a stronger influence when its variance is estimated to be small (for details of the procedure, see (Wiecki et al., 2013)).”

*Do AIC or BIC provide for converging conclusions?*

According to our understanding of the literature on this topic, AIC and BIC are not appropriate for hierarchical models. We now write in the Materials and methods section “Analysis and modeling of choice behavior”:

“To test the robustness of the significance of the TPR-dependent effect on drift criterion, we re-fitted the model, but now fixing drift criterion with TPR, while still allowing all other of the above parameters to vary with TPR. Using deviance information criterion (Spiegelhalter et al., 2002) for model selection, we compared whether the added complexity of our original model was justified to account for the data. This is a common metric for comparing hierarchical models, for which a unique “likelihood” is not defined, and the effective number of degrees of freedom is often unclear (Spiegelhalter et al., 2002).